# Analysis of growing season drought characteristics and driving factors for vegetation in the Santun River Irrigation Area in Xinjiang

Yuxin Wei[1,2], Hongfei Tao[1,2], Yan Xu[3], Mahemujiang Aihemaiti[1,2], Chunlei Lu[4], Youwei Jiang[1,2], Qiao Li[1,2]*

1 College of Hydraulic and Civil Engineering, Xinjiang Agricultural University, Urumqi, China, 2 Xinjiang Key Laboratory of Hydraulic Engineering Security and Water Disasters Prevention, Urumqi, China, 3 Xinjiang Cold and Arid Regions Water Resources and Ecological Water Conservancy Engineering Research Center (Academician Expert Workstation), Urumqi, China, 4 Changji Water Conservancy Management Station (Santun River Basin Management Office), Changji,China

* 931025995@qq.com

## Abstract

Global warming is exacerbating the occurrence of droughts, which have a significant impact on society. Drought is one of the main factors limiting the development of the Santun River Irrigation Area in Xinjiang. Clarifying the driving mechanism and spatial and temporal evolution characteristics of drought in this irrigation area is crucial for ensuring the sustainable development of agriculture. In this paper, the temperature vegetation drought index (TVDI) is used as a drought indicator to analyze the spatial and temporal evolution characteristics of drought in the Santun River Irrigation Area in Xinjiang, as well as to reveal the factors influencing drought using a Geoprobe model. The results show that the mean value of the TVDI in the Xinjiang Santun River Irrigation Area during 19 years was 0. 738, categorizing it as medium drought. During this period, there was an increasing trend of drought in spring and autumn and a decreasing trend of drought in summer. The drought in the irrigation district had strong spatial heterogeneity, and overall, the drought was stronger in the northern part of the region than in the southern part of the region. Over the past 19 years, the light drought areas in the irrigation district shifted to the medium and severe drought classes at a rate of 114.9 km$^2$·10a$^{-1}$. The combined effect of elevation and temperature had the strongest explanatory power for drought occurrence in the irrigated area, with a q-value of 0.869. The results of this study provide a theoretical basis for drought risk assessment and water resource planning in arid regions, as well as a reference for drought monitoring studies in similar regions.

## 1. Introduction

Due to the combination of various factors, including global warming and human activities, drought disasters are becoming more frequent, causing tens of billions of

**Data availability statement:** All relevant data are within the manuscript and its Supporting Information files.

**Funding:** This research was funded by Major Project of Xinjiang Uygur Autonomous Region (2023A02002-1),National Natural Science Foundation of China(41762018), Open Project of Xinjiang Key Laboratory of Water Conservancy Engineering Safety and Water Disaster Prevention (ZDSYS-JS-2021-09), 2023 Research project of Xinjiang Key Laboratory of Water Conservancy Engineering Safety and Water Disaster Prevention (ZDSYS-YJS-2023-10) and The Belt and Road Special Foundation of the National Key Laboratory of Water Disaster Prevention (2020491611).

**Competing interests:** The authors have declared that no competing interests exist.

dollars in economic losses and seriously threatening human society [1,2]. Drought has become one of the most widespread and destructive natural disasters worldwide [3], and it is characterized by the following features: a slow onset and long duration, wide-ranging impacts, significant cumulative effects, and destructive effects on ecosystems and human society [4]. In addition, drought events are characterized by significant spatial and temporal heterogeneity, and their impacts accumulate over time and persist in the ecosystem for a long period of time even after the drought is over [5,6]. Therefore, quickly attaining an accurate understanding of the drought situation is of great significance in guiding the ecological restoration and agricultural production in a region [7].

Traditional drought monitoring mainly reflects regional drought conditions by means of climate data or soil moisture information for specific locations [8]. Many different drought indicators have been proposed to monitor changes in drought severity. The most commonly used indicators include the Palmer drought severity index (PDSI) [9], standardized precipitation index(SPI) [10], and standardized precipitation evapotranspiration index (SPEI) [11]. However, due to differences in global climate conditions, the results of the drought index assessment will vary from region to region [12]. The PDSI lacks the ability to characterize multi-scale drought, and its parameters are uncertain [13]. The SPI only considers the precipitation factor, and it ignores other important factors affecting drought such as evapotranspiration [14]. The SPEI considers the water balance but does not consider soil water information [14].

Remote sensing technology has gradually become a research hotspot because it provides multi-spatiotemporal scale vegetation and surface temperature information for the study of land surface processes, realizes wide-area dynamic monitoring, and greatly improves the efficiency of drought assessment [15,16]. Many scholars have conducted research on the use of remote sensing technology to monitor drought disasters. For example, Yue et al. [17] found that the Climate Forecast System Model (CFSMP) has the potential to mitigate seasonal drought in South China and can be applied to similar regions with similar resource crises. Yang et al. [18] used the SPEI to analyze the temporal and spatial patterns of meteorological drought on the North China Plain from 1970 to 2020 and explored the contributions of climate factors on annual and seasonal scales. Zhang et al. [19] analyzed the spatial and temporal characteristics of drought in Asian grassland ecosystems and the change trend from 2010 to 2018 using the temperature vegetation drought index (TVDI).

At the driver level, droughts occur as a result of the non-linear coupling of natural factors and human activities [20]. Climate warming alters precipitation patterns and increases evapotranspiration from vegetation, leading to more frequent droughts, while human activities alter the water cycle and vegetation cover, further amplifying the effects of drought [21]. Understanding these climate phenomena and the mechanisms by which human activities respond to drought is essential for monitoring and responding to future drought risks. As drought research continues to deepen, scholars around the globe have conducted extensive research on drought-driven mechanisms. For example, Wang et al. [22] revealed that the main drivers of drought in the Yellow River Basin in China are climate and elevation. Gebremichael et al. [23]

found that the main drivers of drought in the Upper Awash Basin in Ethiopia are land use changes and vegetation cover changes. Zhu et al. [24] found that the increase in the water vapor pressure deficit is one of the main drivers of drought propagation in Central Asia.

Although many studies have explored the spatial and temporal changes in drought and its influencing factors, there are some shortcomings in existing studies. First, most of the drought monitoring studies focused on large-scale regional analysis, and there is a lack fine monitoring of ecologically sensitive units such as typical irrigation zones over long time scales. Second, most of the existing studies focused on drought inversion and did not reveal enough about the coupling mechanism between drought and natural and human activities. Finally, it is often difficult for traditional research methods to quantify both the spatial distribution characteristics and temporal changes of drought, thus hindering the attainment of a deeper understanding of the evolution and driving mechanisms of drought. In order to more comprehensively analyze the spatial and temporal characteristics of drought and its driving mechanisms, the combination of Geodetector and trend analysis has become an innovative approach for studying drought dynamics [25].

The Xinjiang Santun River Irrigation Area, as the core of the economic belt on the north slope of the Tianshan Mountains in Xinjiang, China, is the solid foundation of Changji's agricultural economy and plays a vital role in ensuring the stable development of the city's economy. Since 2005, a series of water resource management and ecological protection measures have been taken in the Santun River Irrigation Area, Xinjiang. However, there are still two gaps in research on drought in this region. (1) There is a lack of quantitative analysis of the spatial and temporal heterogeneity of drought in a long time series (>15 years). (2) The interactive driving mechanism of multi-dimensional factors such as the topographic gradient (elevation and slope), climate change (temperature) and human activities (gross domestic product (GDP) and land use types) on the occurrence of drought has not yet been explained. These research gaps directly restrict scientific decision-making for the optimal allocation of water resources and drought prevention and control in irrigated areas. Clearly understanding the spatiotemporal differentiation and driving mechanism of drought in this region can provide a scientific basis for optimizing the irrigation system and planting structure.

In view of this, the two research objectives of this study were (1) to reveal the drought intensity and spatial-temporal differentiation patterns in the Santun River Irrigation Area based on Landsat (thematic mapper (TM)/enhanced thematic mapper plus (ETM+)/ operational land imager (OLI)_thermal infrared sensor (TIRS)) time-series datasets for 2005–2023 by integrating trend analysis and spatial transfer matrix methods through the TVDI model; and (2) to quantify the contributions of topographic factors, climatic factors, and anthropogenic factors to the formation of the spatial pattern of drought and their interaction effect. In this study, an entire chain of drought monitoring-driver analysis-strategy response was realized. The results of this study provide strong support for the optimal allocation of water resources, adjustment of the irrigation system and policy formulation, and promotion of sustainable regional development.

## 2. Study area

### 2.1 Study area

The Xinjiang Santun River Irrigation Area (86°24'–87°37'E, 43°26'–45°20'N) is located in the Xinjiang Changji Hui Autonomous Prefecture, in the northern foothills of the Tianshan Mountains, and at the southern edge of the Junggar Basin (Fig 1). The main water source projects in this area are the Santun River Reservoir and Nurga Reservoir, and the backbone water transmission projects are the east and west trunk canals, making this a typical oasis agricultural system in an arid region. As a large irrigation area in China, the Santun River Irrigation Area is responsible for irrigating 680 km² of farmland. The irrigation area has an elevations of 415–1315 m, a total area of about 730 km², and a total population of about 403,500 people. The entire area exhibits an irregular cluster shape. Subject to its geographic

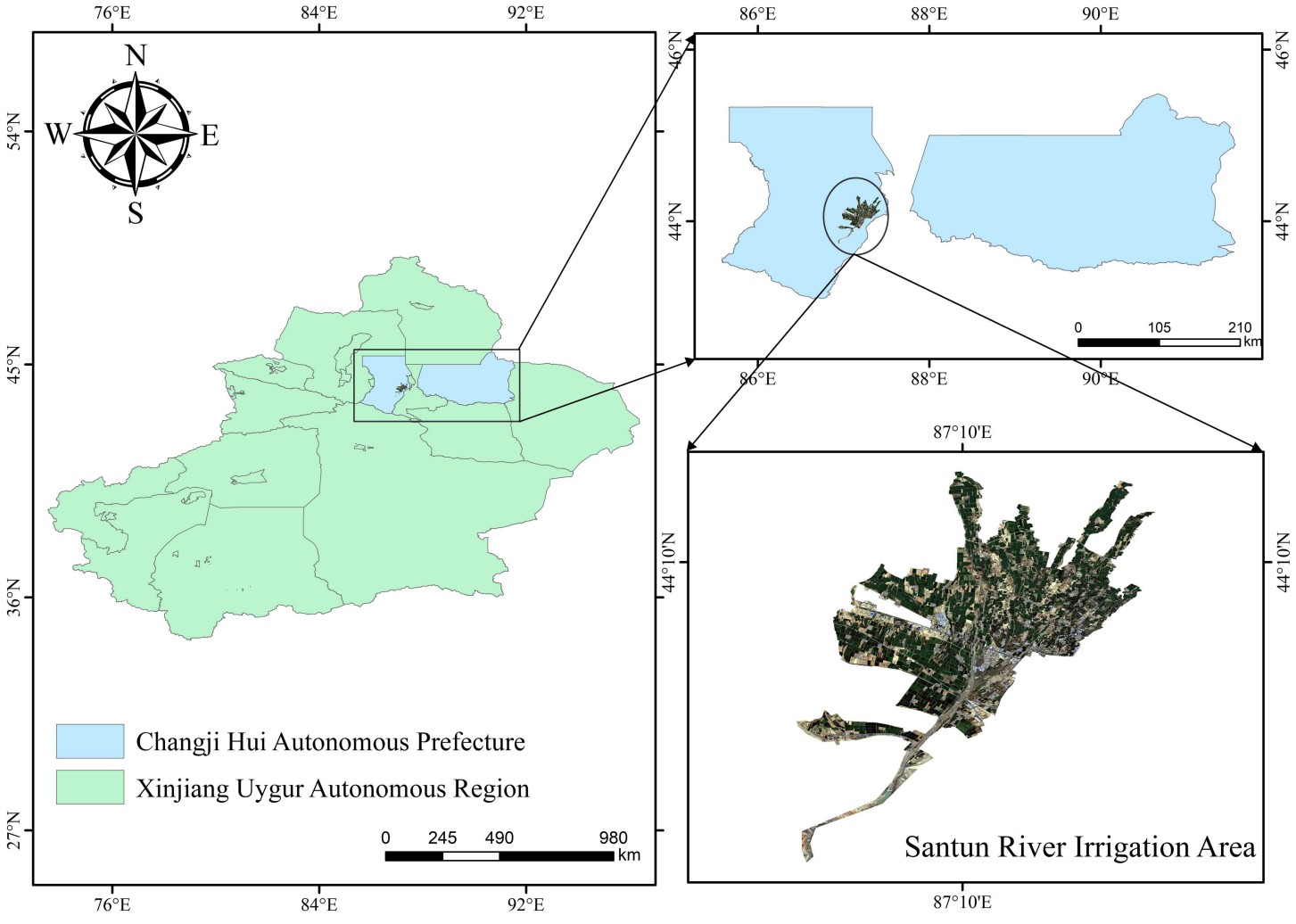

**Fig 1. Study area.**

location and topographic conditions, the climate in the study area is complex, with an average annual temperature of 6.87°C and precipitation of 280.03 mm. The overall climate features include severe winters, hot summers, rare rainfall, strong evaporation, and a large temperature difference between day and night. The crops in the irrigated areas mainly include wheat, corn, cotton, and other crops, and drought is the dominant factor restricting local development.

## 2.2 Materials and methods

Considering the influences of the weather, cloud cover, and seasonal vegetation cover in the study area [26,27], in this study, Landsat TM/ETM＋/OLI_TIRS series remote sensing images (strip number: 143, line number: 29, 30, resolution: 30 m for period 114 from 2005 to 2023 with clear weather and cloud cover below 10%,) were selected for analysis. ENVI software was utilized to conduct remote sensing image geometric correction, stripping, radiometric calibration, atmospheric correction, mosaicking, cropping, and mask pre-processing operations [28]. Additional data sources are presented in Table 1. in the remote sensing image information is presented Table 2.

**Table 1. Data information.**

| Data | Resolution (m) | Data availability | Data sources |
|---|---|---|---|
| Landsat remote sensing data | 30 | 2005–2023 | United States Geological Survey website (http://glovis.usgs.gov/)<br>China National Geospatial Data Cloud (http://www.gscloud.cn/) |
| Digital elevation model | 30 | 2005–2023 | China National Geospatial Data Cloud (http://www.gscloud.cn/) |
| Slope | 30 | 2005–2023 | China National Geospatial Data Cloud (http://www.gscloud.cn/) |
| Aspect | 30 | 2005–2023 | China National Geospatial Data Cloud (http://www.gscloud.cn/) |
| Temperature | / | 2005–2023 | China National Meteorological Information Center (http://data.cma.cn/) |
| Precipitation | / | 2005–2023 | China National Meteorological Information Center (http://data.cma.cn/) |
| Gross domestic product | 1000 | 2005–2023 | Data Center for Resource and Environmental Sciences, Chinese Academy of Sciences (http://www.resdc.cn/) |
| Land use type | 30 | 2005–2023 | Data Center for Resource and Environmental Sciences, Chinese Academy of Sciences (http://www.resdc.cn/) |
| Agrotype | 1000 | 2005–2023 | Data Center for Resource and Environmental Sciences, Chinese Academy of Sciences (http://www.resdc.cn/) |
| Measured soil moisture | / | 2017 | Deployment of Smart Moisture Monitors in Santun River Irrigation Area, Xinjiang |

## 2.3 Methods of data analysis

Xinjiang is the core of the Silk Road Economic Belt, but its special geographic environment has led to the frequent occurrence of drought events. Previous studies on drought monitoring in Xinjiang have mostly discussed the influences of meteorological factors on drought in large-scale regions. In this study, in addition to using Landsat series satellite data to calculate the TVDI index, the potential influences of factors such as the topography, geomorphology, land use types, gross domestic product (GDP), and human activities on drought were analyzed. The specific workflow of the research is illustrated in Fig 2.

**2.3.1 Calculation of temperature–vegetation drought index based on Ts-NDVI feature space.** Among the many drought indices developed, compared with drought indices such as the SPI and SPEI, the TVDI has a clear physical mechanism for retrieving drought through the land surface temperature (LST)–normalized difference vegetation index (NDVI) feature space, which can effectively eliminate the influence of a single factor on monitoring results. The model is simple and effective, which is more suitable for agro-ecosystem monitoring [29]. According to relevant studies, the LST and NDVI exhibit an obvious negative correlation at different temporal and spatial resolutions [30]. The theoretical value range of the TVDI is (0,1). The higher the value, the drier the representative pixel that represents the region. The specific calculation formulas are as follows:

$$TVDI = \frac{T_s - T_{s\_min}}{T_{s\_max} - T_{s\_min}},$$

(1)

$$T_{s\_max} = a_{max} + b_{max} NDVI,$$

(2)

$$T_{s\_min} = a_{min} + b_{min} NDVI,$$

(3)

where Ts_max is the maximum surface temperature, constituting the dry edge of the feature space; Ts_min is the minimum surface temperature, constituting the wet edge of the feature space; $a_{max}$ and $b_{max}$ are the dry-side linear fit parameters; and $a_{min}$ and $b_{min}$ are the wet-side linear fit parameters.

**Table 2. Remote sensing imagery schedule.**

| Number | Date | Number | Date | Number | Date |
|---|---|---|---|---|---|
| 1 | 8-Apr-2005 | 39 | 15-Aug-2011 | 77 | 22-July-2017 |
| 2 | 24-Apr-2005 | 40 | 31-Aug-2011 | 78 | 7-Aug-2017 |
| 3 | 14-Aug-2005 | 41 | 18-Oct-2011 | 79 | 8-Sept-2017 |
| 4 | 30-Aug-2005 | 42 | 3-Nov-2011 | 80 | 12-Oct-2017 |
| 5 | 1-Oct-2005 | 43 | 11-Apr-2012 | 81 | 10-Aug-2018 |
| 6 | 17-Oct-2005 | 44 | 24-Apr-2012 | 82 | 26-Aug-2018 |
| 7 | 13-May-2006 | 45 | 17-Aug-2012 | 83 | 9-Oct-2018 |
| 8 | 29-May-2006 | 46 | 2-Sep-2012 | 84 | 25-Oct-2018 |
| 9 | 1-Aug-2006 | 47 | 4-Oct-2012 | 85 | 9-May-2019 |
| 10 | 17-Aug-2006 | 48 | 20-Oct-2012 | 86 | 25-May-2019 |
| 11 | 20-Oct-2006 | 49 | 22-Apr-2013 | 87 | 13-Aug-2019 |
| 12 | 5-Nov-2006 | 50 | 8-May-2013 | 88 | 29-Aug-2019 |
| 13 | 22-Apr-2007 | 51 | 12-Aug-2013 | 89 | 16-Oct-2019 |
| 14 | 8-May-2007 | 52 | 28-Aug-2013 | 90 | 1-Nov-2019 |
| 15 | 20-Aug-2007 | 53 | 15-Oct-2013 | 91 | 11-May-2020 |
| 16 | 5-Sept-2007 | 54 | 31-Oct-2013 | 92 | 27-May-2020 |
| 17 | 8-Nov-2007 | 55 | 9-Apr-2014 | 93 | 15-Aug-2020 |
| 18 | 24-Nov-2007 | 56 | 25-Apr-2014 | 94 | 31-Aug-2020 |
| 19 | 31-Mar-2008 | 57 | 15-Aug-2014 | 95 | 2-Oct-2020 |
| 20 | 16-Apr-2008 | 58 | 31-Aug-2014 | 96 | 18-Oct-2020 |
| 21 | 6-Aug-2008 | 59 | 12-Oct-2014 | 97 | 14-May-2021 |
| 22 | 22-Aug-2008 | 60 | 28-Oct-2014 | 98 | 30-May-2021 |
| 23 | 9-Oct-2008 | 61 | 12-Apr-2015 | 99 | 2-Aug-2021 |
| 24 | 25-Oct-2008 | 62 | 28-Apr-2015 | 100 | 18-Aug-2021 |
| 25 | 3-Apr-2009 | 63 | 6-July-2015 | 101 | 5-Oct-2021 |
| 26 | 19-Apr-2009 | 64 | 22-July-2015 | 102 | 21-Oct-2021 |
| 27 | 22-June-2009 | 65 | 5-Oct-2015 | 103 | 1-May-2022 |
| 28 | 8-July-2009 | 66 | 21-Oct-2015 | 104 | 17-May-2022 |
| 29 | 26-Sept-2009 | 67 | 16-May-2016 | 105 | 5-Aug-2022 |
| 30 | 12-Oct-2009 | 68 | 30-Apr-2016 | 106 | 21-Aug-2022 |
| 31 | 8-May-2010 | 69 | 4-Aug-2016 | 107 | 8-Oct-2022 |
| 32 | 24-May-2010 | 70 | 20-Aug-2016 | 108 | 24-Oct-2022 |
| 33 | 9-June-2010 | 71 | 7-Oct-2016 | 109 | 25-Mar-2023 |
| 34 | 25-June-2010 | 72 | 23-Oct-2016 | 110 | 10-Apr-2023 |
| 35 | 15-Oct-2010 | 73 | 3-May-2017 | 111 | 16-Aug-2023 |
| 36 | 31-Oct-2010 | 74 | 19-May-2017 | 112 | 1-Sept-2023 |
| 37 | 9-Apr-2011 | 75 | 6-June-2017 | 113 | 3-Oct-2023 |
| 38 | 25-Apr-2011 | 76 | 22-June-2017 | 114 | 19-Oct-2023 |

Annotation: For 2005–2012, the remote sensing image sensor model is Landsat 7 ETM; for 2013–2021 and 2023, the remote sensing image sensor model is Landsat 8 OLI; and for 2022, the sensor model is Landsat 9 OLI.

In this study, the drought in the Santun River Irrigation Area was divided into five levels according to relevant research results on the temperature vegetation drought index, as well as the actual soil water content in the study area and the drought levels classification methods utilized in previous studies [31–33]. The details are presented in Table 3.

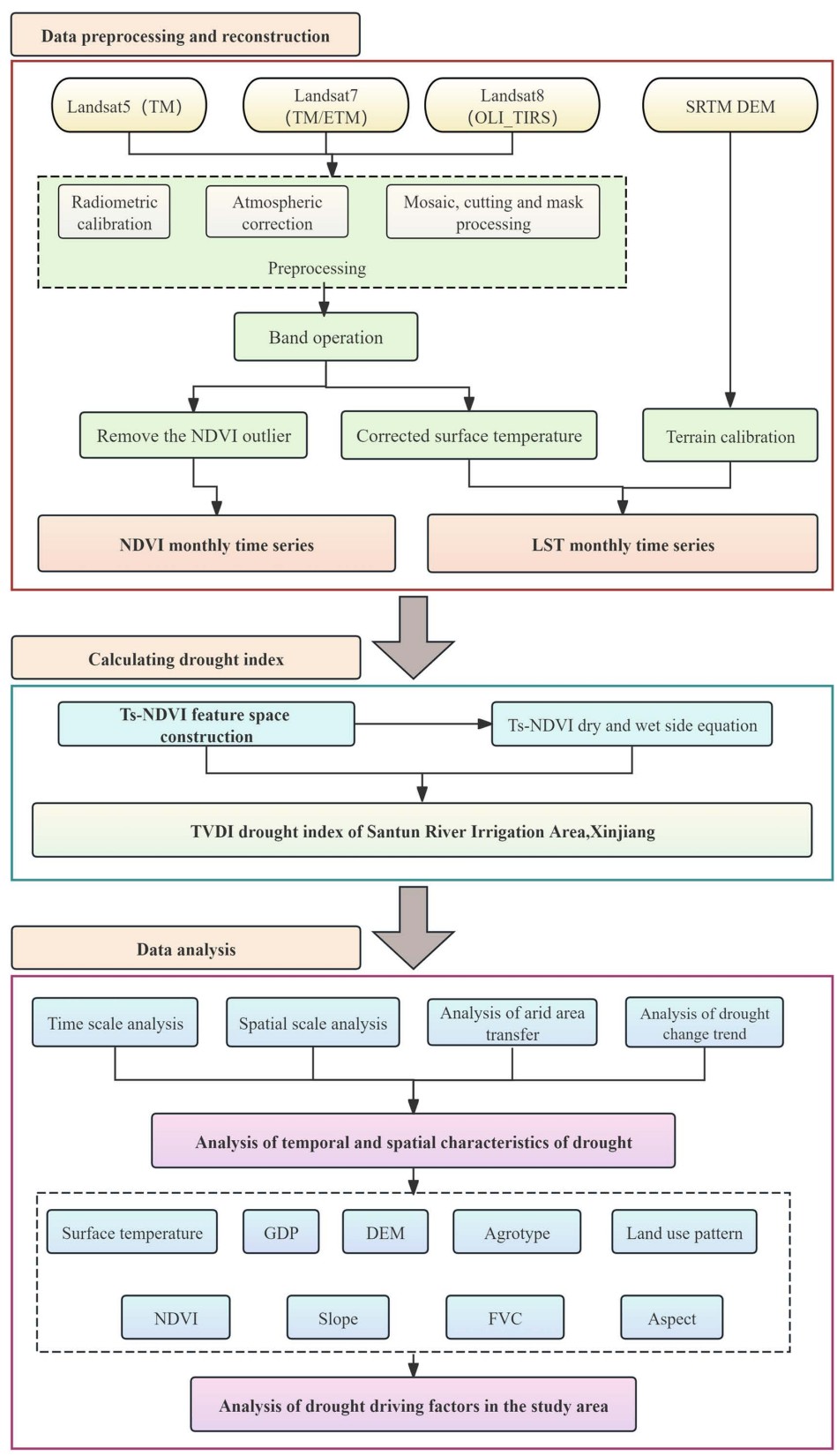

**Fig 2. Workflow of the study.**

**Table 3. TVDI drought monitoring grade classification standard in the Santun River Irrigation Area.**

| Range of TVDI values | Degree of drought | Drought expression |
|---|---|---|
| 0–0.6 | No drought | Normal vegetation development |
| 0.6–0.71 | Mild drought | The air near the surface is dry |
| 0.71–0.76 | Moderate drought | The leaves of the plants wilted |
| 0.76–0.85 | Severe drought | The soil appears to be thick and dry |
| 0.85–1.0 | Extreme drought | The soil cracked and the vegetation died |

**2.3.2 Drought trend analysis under time series.** In this study, Theil-Sen trend analysis and the Mann-Kendall (MK) test were used to analyze the change trend of the drought. These methods do not depend on the specific distribution form of the data, have a strong tolerance for errors in the data, can handle the discontinuity problem of remote sensing data without interpolation, and can complementarily analyze the spatiotemporal evolution characteristics of drought, thereby providing a robust analysis framework for drought research [34,35]. The formula is as follows:

$$\beta = Median \left\{ \frac{(x_j - x_i)}{j - i} \right\} \qquad \forall j > i,$$

(4)

where $\beta$ is the median of the data pairs' slopes; Median () is the functions used to take the median of a data pair; $x_i$ is item i of the raster data in the time series; and $x_j$ is item j of the raster data in the time series:

$$S = \sum_{i=1}^{n-1} \sum_{j=i+1}^{n} sgn(x_j - x_i) \qquad i < j \leq n,$$

(5)

where sgn() is a symbolic function. The specific calculation formula is as follows:

$$sgn(x_j - x_i) = \begin{cases} 1 & x_j - x_i > 0 \\ 0 & x_j - x_i = 0 \\ -1 & x_j - x_i < 0 \end{cases},$$

(6)

where S is the statistic representing the test; and n is the sample size of the time series. When n < 10, S is used directly for trend testing; and when n > 10, S is standardized and is later converted to the test statistic Z.

The specific formula for calculating the statistic Z is as follows:

$$Z = \begin{cases} \frac{S-1}{\sqrt{var(S)}} & S > 0 \\ 0 & S = 0 \\ \frac{S+1}{\sqrt{var(S)}} & S < 0 \end{cases},$$

(7)

$$var(S) = \frac{n(n-1)(2n+5) - \sum_{i=1}^{m} t_i (t_i - 1)(2t_i + 5)}{18},$$

(8)

where $\sqrt{var(S)}$ is the variance of S; n is the total length of the sample year; m is the number of recurring datasets in the sequence; and $t_i$ is the number of duplicates in the ith duplicate dataset.

If Z > 0, the trend of TVDI is increasing, and vice versa. If Z = 0, TVDI does not exhibit a significant trend. In this study, the significance level is α=0.05. When |Z| is greater than 1.96 and 2.58, the trend passes the significance test at the 95%

and 99% confidence levels, respectively. The criteria for determining the significance of a specific trend are shown in Table 4.

**2.3.3 Spatial transition matrix.** The space transition matrix, also known as the transition matrix or state transition matrix, is mainly used to analyze and predict the changes between different states of the system. It can analyze complex transformations into basic operations and is an important mathematical tool for describing the state transition law of a system in discrete time [36]. By constructing and applying the spatial transfer matrix, in this study, the transfer conditions and rules of the different drought grades in the study area were investigated. The specific formula is as follows:

$$S_{ij} = \begin{pmatrix} S_{11} & \dots & S_{1n} \\ \vdots & \ddots & \vdots \\ S_{n1} & \dots & S_{nn} \end{pmatrix},$$

(9)

where $S_{ij}$ is the area of class i data converted to class j; and n is the total number of graded area divisions

**2.3.4 Geodetector model.** The selection of the Geodetector for use in this study was based on the advantages of its method and the high adaptability of the drought driving mechanism; that is, as a non-parametric chemical tool, it does not need to presuppose the mathematical relationship between variables, effectively avoids the bias of the traditional regression model caused by function mis-setting, and is suitable for nonlinear or threshold response analysis between drought factors and indexes [37]. In this study, the TVDI was taken as the dependent variable, and nine types of factors, including natural factors and human factors, were taken as the independent variables. Through the detection module of single factor and double factor interaction, the drought driving factors of the TVDI in the Santun River Irrigation District and the influence of each factor interaction on the TVDI were studied. The specific calculation formula is as follows:

$$q = 1 - \frac{\sum_{h=1}^{L} N_h \sigma_h^2}{\sigma^2 N^2} = 1 - \frac{SSW}{SST},$$

(10)

where q is the explanatory power of each influencing factor for TVDI; h is the number of the stratification of the influencing factors for each variable; Nh and N are the number of units in layer h and the total number of units in the study area, respectively; $\sigma^2$ is the variance and global variance of the different graded regions; SSW is the sum of the intra-layer variances; and SST is the total variance.

The nine driving factors considered in this study were the temperature, slope, slope direction, elevation, soil type, GDP, NDVI, land use type, and fractional vegetation cover (FVC). The driving logic of each driving factor for the occurrence of drought disaster is presented in Table 5.

In this study, in the ArcGIS software, the natural breakpoint method was used to reclassify the temperature, slope, slope direction, soil type, GDP, NVI, and vegetation cover indicators, each of which was divided into 10 categories, and

**Table 4. Mann–Kendall trend test grading.**

| β | |Z| | Trend classification |
|---|---|---|
| β>0 | 2.58<|Z| | Extremely significant desiccation |
| | 1.96<|Z|≤2.58 | Significant desiccation |
| | 1.96≥|Z| | Slight desiccation |
| β | |Z|=0 | Intact |
| β<0 | 1.96≥|Z| | Mild relief |
| | 1.96<|Z|≤2.58 | Significant relief |
| | 2.58<|Z| | Extremely significant relief |

**Table 5. The driving logic of drought by the different driving factors.**

| Driving factor | Driver logic |
|---|---|
| Temperature | Temperature directly affects the soil water deficit by regulating surface evapotranspiration, which leads to drought [38]. |
| Slope | The slope angle controls the surface runoff rate and water retention capacity and induces drought [39]. |
| Aspect | The slope aspect affects the surface evapotranspiration and induces drought through the difference in the solar radiation [40]. |
| DEM | The water-heat balance is regulated by the vertical differentiation of the temperature lapse rate (0.6°C/100 m) and precipitation [40]. |
| Soil type | The water holding capacities of different soil types are different, which directly affects the threshold of drought occurrence [40]. |
| NDVI | Reflects the vegetation greenness, biomass, and photosynthetic activity intensity [41] |
| FVC | Quantifies the spatial density of the vegetation cover (coverage ratio) [41] |
| GDP | The intensity of the regional economic development breaks the balance of the regional water cycle and induces drought [42]. |
| Land use type | Land use type changes alter the regional water use intensity and induce drought [43]. |

the land use type indicators were divided into six categories. After this, a fishing net was constructed based on the specification of 500 m × 500 m in the Santun River Irrigation Area, and it was clipped based on the vector map of the irrigation area to obtain a vector map of the irrigation area covered by the fishing net. Using the interactive detection module of Geodetector, the attribute values of the nine indicators were extracted for data analysis, and the center of each grid was taken as a sampling point. The classification scheme is presented in Table 6.

## 3. Results

### 3.1 Evaluation of the credibility of the drought monitoring indicators for the vegetation growing season in the Santun River Irrigation Area, Xinjiang

According to the physical meaning of the TVDI, the TVDI value is negatively correlated with the soil moisture. When the soil water content of the 0–20 cm layer is used as the validation standard for the drought classification, the evaluation results are more accurate. A large number of related studies have verified this conclusion [44]. In this study, the soil moisture content at a depth of 10 cm was selected to evaluate the credibility of the TVDI monitoring results. Information about the spatial extent of the measured soil moisture content sampling points is shown in Fig 3. Due to the long study period, the missing soil moisture data, and the large amount of month-by-month validation data, in this study, the measured soil water content data for the irrigation area in May, June, July, August, and September 2017 corresponding to the TVDI obtained via inversion of the remote sensing image data for the same period were selected for use in conducting the correlation analysis and validation. As shown in Fig 4, the overall fitting coefficient of determination $R^2$ of the TVDI and the measured soil water content for each period in the irrigation area was 0.47, and the Pearson correlation coefficient was –0.721, which all passed the significance test (P = 0.05). Overall, the TVDI exhibited a significant negative correlation with the soil water content, so the TVDI calculated from the Landsat series remote sensing data for drought monitoring in the Santun River Irrigation Area in Xinjiang is highly credible.

**Table 6. Classification of TVDI effector interactions.**

| Basis of judgment | Type of interaction |
|---|---|
| $q(X1 \cap X2) < min(q(X1), q(X2))$ | Nonlinear weakening |
| $min(q(X1), q(X2)) < q(X \cap X2) < max(q(X1), q(X2))$ | Single-factor nonlinear attenuation |
| $q(X1 \cap X2) < max(q(X1), q(X2))$ | Two-factor enhancement |
| $q(X1 \cap X2) = q(X1) + q(X2)$ | Separate |
| $q(X1 \cap X2) > q(X1) + q(X2)$ | Nonlinear enhancement |

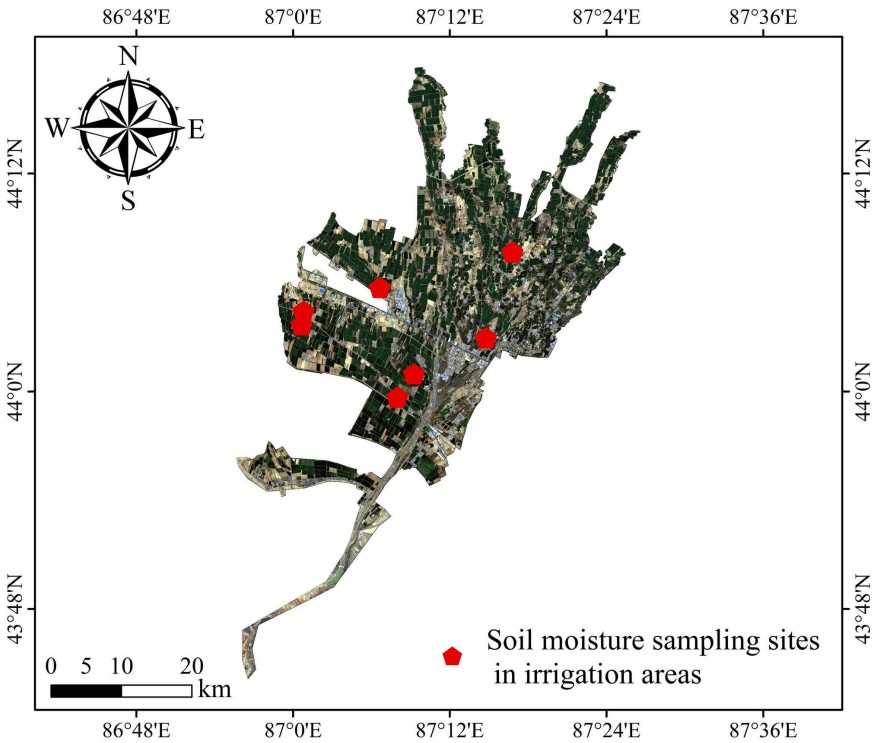

**Fig 3. Distribution of sampling sites in the study area.**

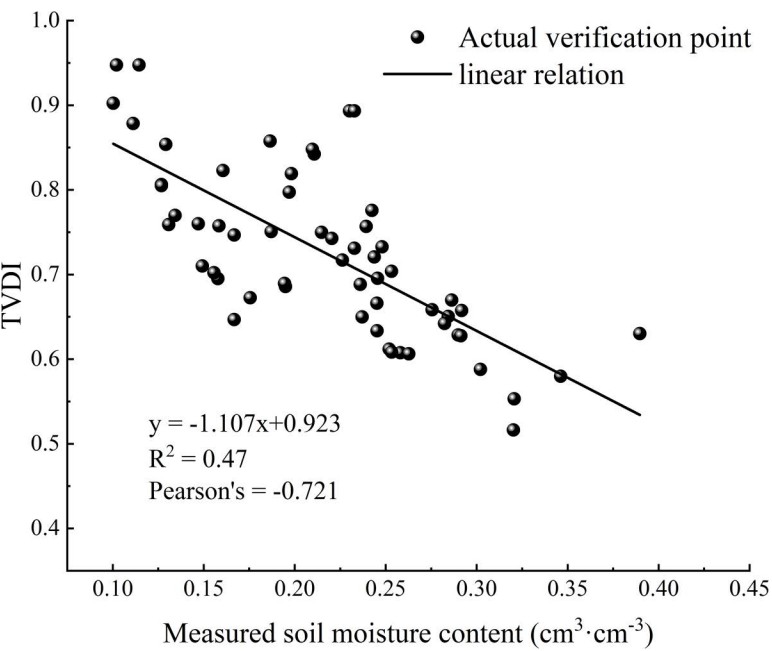

**Fig 4. Correlation between soil moisture content and TVDI.**

## 3.2  Construction of Ts-NDVI feature space and fitting of the wet and dry edges

Fig 5 shows the results of the dry and wet edge fitting of the Ts-NDVI feature space for the Santun River Irrigation Area. The sampling interval is 10 image points for each period due to the large amount of data, and it is illustrated using the data for the following eight periods as an example. As can be seen from Fig 5, the slope of the dry edge was negative for many years, and the mean $R^2$ value was as high as 0.9 for many years, indicating that the dry edge fitting effect was excellent. With increasing NDVI, the land surface temperature decreased, and the two had a strong negative correlation. The slope of the wet-edge fitting equation was greater than zero except in 2022, indicating that in most cases, the minimum surface temperature increased when the NDVI increased. The absolute value and $R^2$ value of the fitted equations for the dry side exceeded those for the wet side, i.e., the sensitivity of the fitted maximum surface temperature to the changes in the NDVI was higher for the dry side than for the wet side. Therefore, the dry side had a better overall fit.

## 3.3  Characteristics of the spatial and temporal variations in the TVDI during the vegetation growing season in the Santun River Irrigation Area, Xinjiang

**3.3.1  Trends of the drought duration.**  To analyze the characteristics of the time series changes in the TVDI in the Santun River Irrigation Area in Xinjiang from 2005 to 2023 on different time scales, in this study, the annual average TVDI, the TVDI for the different drought categories, and the percentage of the area during 2005–2023 were statistically analyzed using linear trend analysis (Fig 6). It can be seen that during the past 20 years, the annual average TVDI in the irrigation area varied between 0.699 and 0.774, and the average TVDI was 0.738 in many years. According to the drought classifications presented in Table 2, the drought condition in the irrigation area was medium drought all year. The peak TVDI value occurred in 2016 (0.774), and the lowest value occurred in 2011 (0.699). Thus, the drought condition in the irrigation area as a whole exhibited a fluctuating increasing trend, and the growth rate was 0.0008/a according to the linear fitting.

In addition, the irrigation area in which moderate drought occurred accounted for the largest multi-year proportion, with an average value of 53.01%. The area with severe drought accounted for the next highest proportion (26.52%). The area with mild drought accounted for 20.1%. The areas with no drought or extreme drought were very small, accounting for only 0.2% and 0.16% of the study area. During the last 20 years, the percentage of the area with moderate drought ranged from 19.07% to 68.32%, and it continuously increased at a rate of 0.0019/a. The area with moderate drought accounted for the largest percentage of the total area (498 km$^2$) in 2015. The proportion of the area with severe drought ranged from 1.54% to 77.67%, and it exhibited an increasing trend with a rate of 0.0089/a. The area with severe drought was the largest (about 566 km$^2$) in 2016. The percentage of the area with mild drought ranged from 0.92% to 69.43% during the last 20 years, and it decreased slowly at a rate of –0.0064/a. The proportion of the area with mild drought (506 km$^2$) occurred in 2011. Based on the interannual trends of the temperature, precipitation, and TVDI in the study area (Fig 7), it can be seen that in the study area, the TVDI was positively correlated with the temperature and negatively correlated with the precipitation during the last 20 years. As the temperature increased and the precipitation decreased, droughts occurred frequently in the Santun River Irrigation Area in Xinjiang, and the trend of the droughts gradually increased.

From the perspective of the vegetation growth season, the TVDI exhibited different trends with seasonal changes (Fig 8). Based on the statistical analysis, the mean value of the spring TVDI (March–May) was 0.772, the mean value of the summer TVDI (June–August) was 0.722, and the mean value of the autumn TVDI (September–November) was 0.721. Linear fitting of the TVDI for each season yielded linear trends of 0.0049 a$^{-1}$ (spring), –0.0047 a$^{-1}$ (summer), and 0.0023 a$^{-1}$ (autumn). This indicates that from 2005 to 2023, the drought in the Santun River Irrigation Area of Xinjiang was reduced in the summer and enhanced in the spring and fall.

**3.3.2  Characteristics of the spatial distribution of drought.**  The spatial distribution of the TVDI in the Santun River Irrigation Area in Xinjiang from 2005 to 2023 is shown in Fig 9. As can be seen from Fig 9, the distribution of the

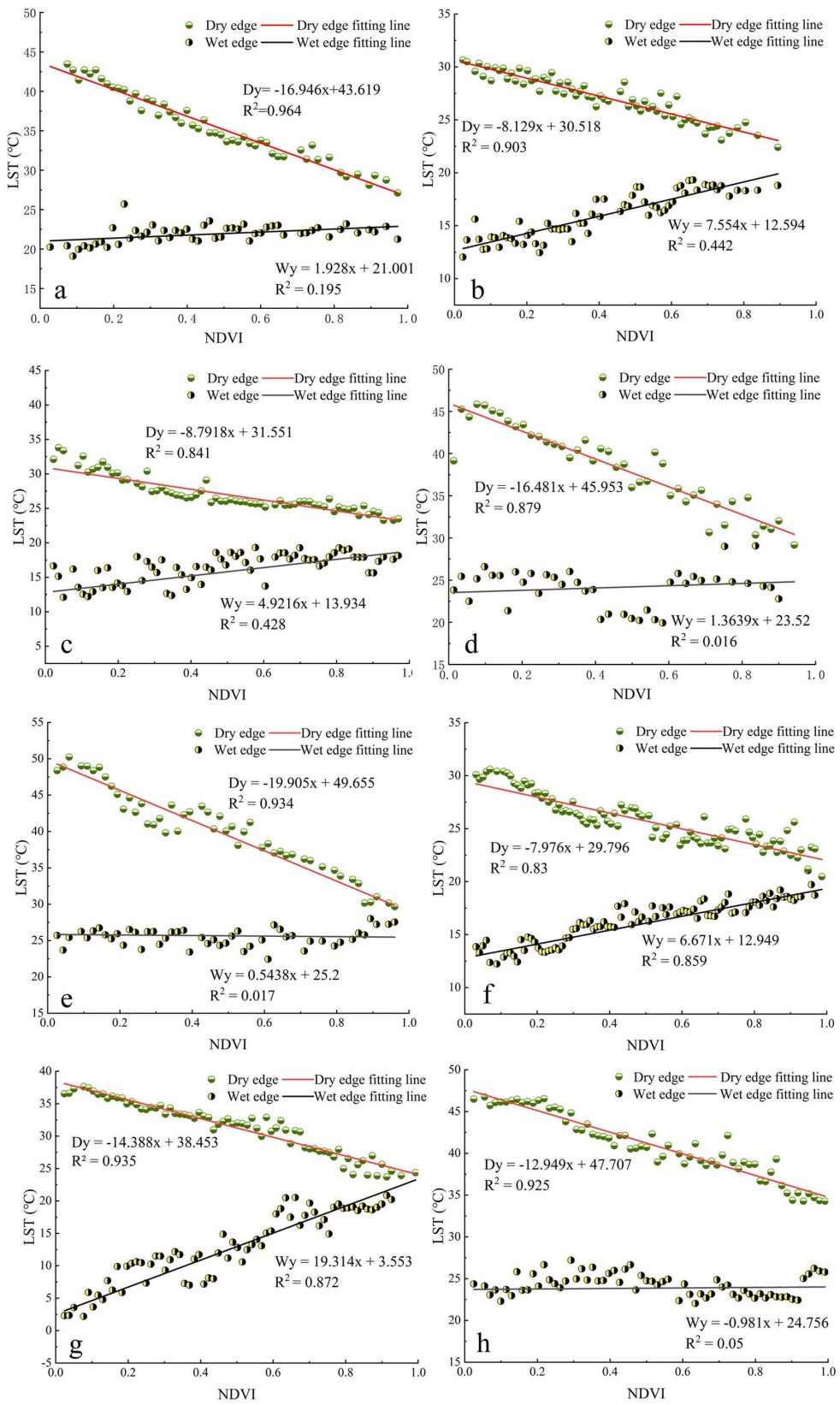

**Fig 5. Spatial dry and wet side fitting of the TS-NDVI characteristics for different time periods in the Santun River Irrigation Area: (a) 2006/05; (b) 2007/04; (c) 2008/10; (d) 2009/08; (e) 2011/08; (f) 2014/04;(g) 2017/05; and (h) 2022/05.**

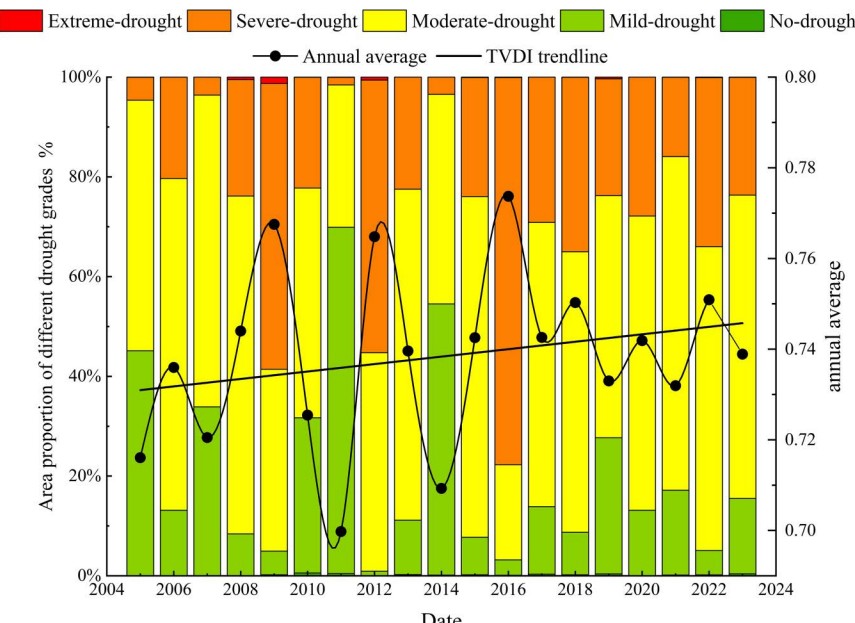

**Fig 6. Inter-annual variations in the TVDI and proportions of the study area with different drought classes in the Santun River Irrigation Area during 2005–2023.**

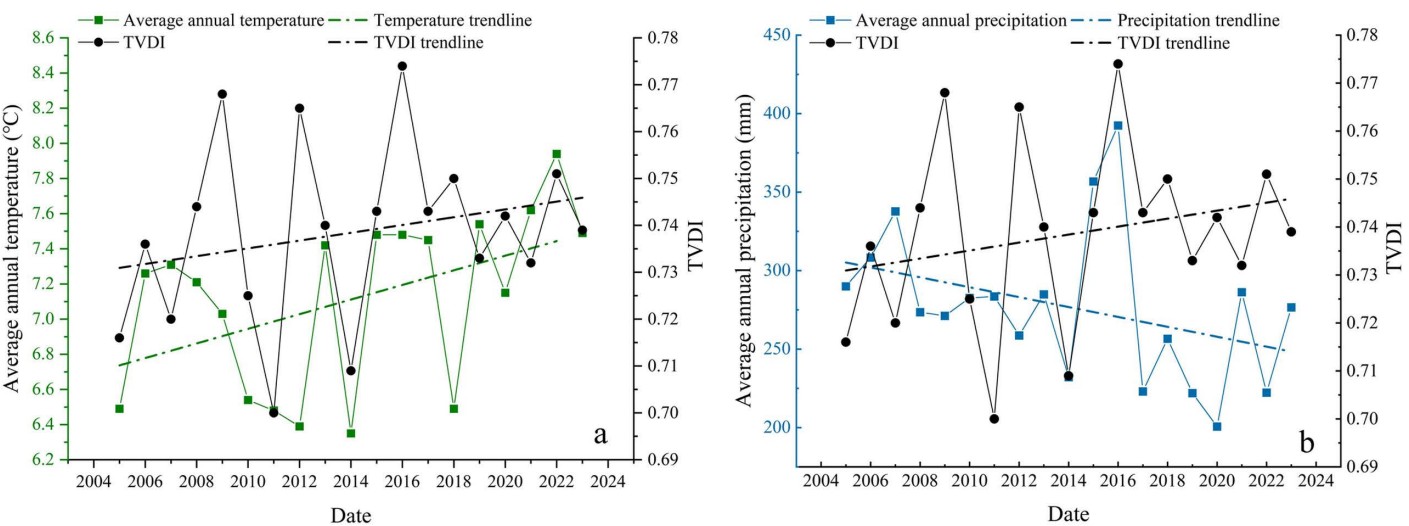

**Fig 7. Trends of the temperature, precipitation, and TVDI in the Santun River Irrigation Area, Xinjiang, during 2005–2023.** (a) Relationship between the mean annual temperature and the TVDI; (b) Relationship between the mean annual precipitation and the TVDI.

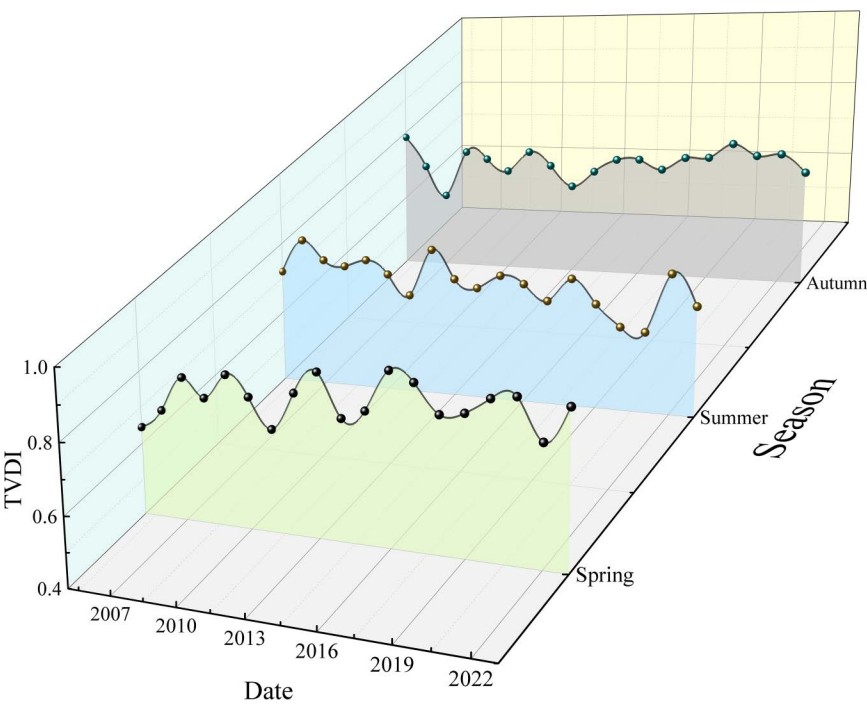

**Fig 8. Temporal trends of the TVDI in different seasons in the Santun River Irrigation Area.**

TVDI in the irrigation district exhibited very strong spatial and temporal heterogeneity, with light drought and moderate drought being the most common levels of drought severity. The drought was weaker in the south and southwest than in the north and northeast. In general, the irrigated areas exhibited a trend of gradual drying during the past 20 years, and the frequency of the occurrence of moderate drought was much higher in the northwest and central areas than in the south. The average annual TVDI in the irrigated area was 0.738, indicating medium drought. The area with a low degree of drought was mainly concentrated in the southern part of the irrigation area. This area was characterized by a relatively high topography, proximity to the water source, i.e., the reservoir, abundant surface water resources, a relatively high vegetation coverage, a TVDI of basically below 0.6, and no drought. Low drought and medium drought were the main types of drought in the irrigated area, which were mainly distributed in the southern and central parts of the irrigated area. In this area, the TVDI was 0.6–0.76. The regions with higher degrees of drought were mainly concentrated in the central and northern parts of the irrigated area, and the TVDI value was higher closer to the northern region.

The spatial distribution of the TVDI in different seasons in the Santun River Irrigation Area exhibited significant differences due to a variety of factors such as seasonal solar radiation, air temperature, rainfall, land use types, geographic location, and vegetation type. As shown in Fig 10, in general, the frequency of drought was greater in the central and northern parts of the irrigation area than in the other areas. In the past 20 years, the irrigation area was mainly dominated by severe drought in spring, and the proportion of the area with drought accounted for as high as 78.44%. The drought was alleviated in summer and fall. The area was mainly dominated by light and medium drought in summer, and the proportions of the area with light and medium drought accounted for 44.14% and 32.75% of the irrigation area, respectively. In the fall, the area was mainly dominated by medium drought, and the area with drought accounted for 80.38%. The areas with TVDI values indicating moderate drought and severe drought were significantly larger in 2009, 2012, and 2016 than in the other years, the proportion of the area with severe drought was greater than 56.31%, and the TVDI of the entire irrigation area exhibited an obvious increasing trend compared with the other years. In 2005, 2011, and 2014, the

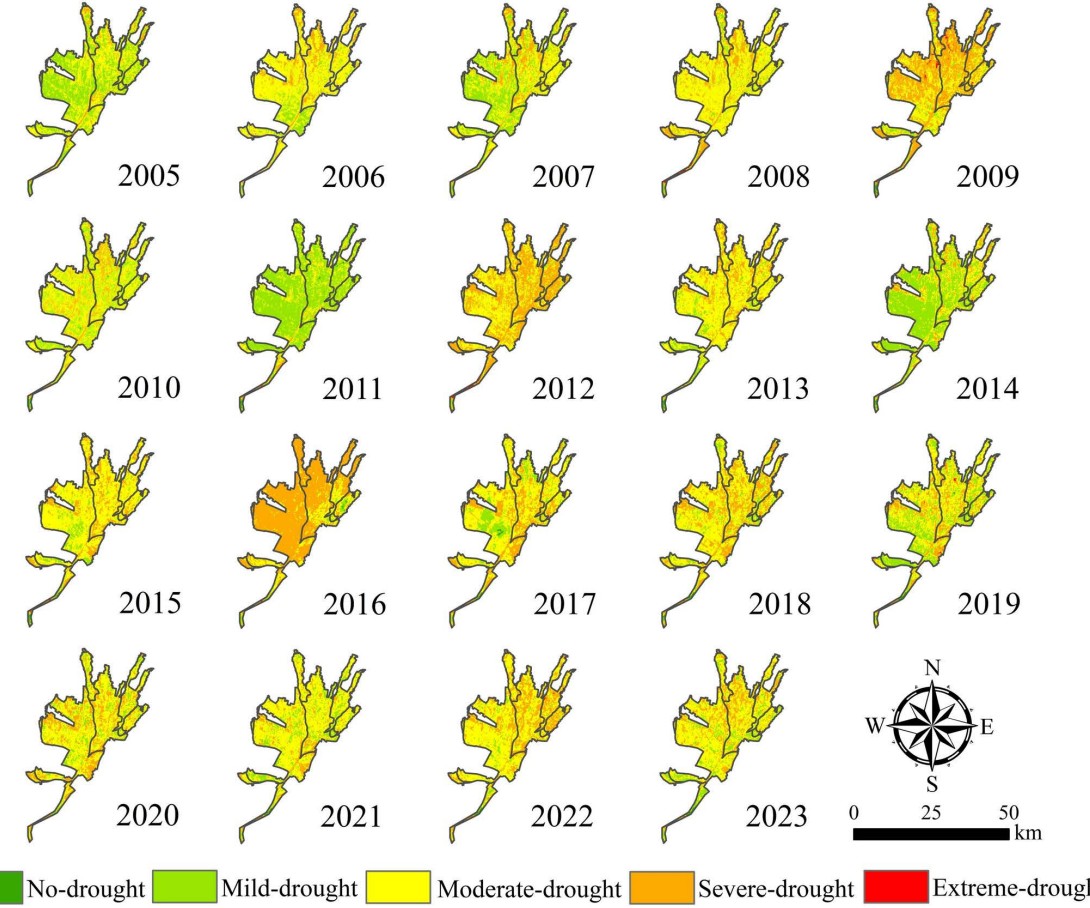

No-drought  Mild-drought  Moderate-drought  Severe-drought  Extreme-drought

**Fig 9. Spatial distributions of the TVDI drought classes in the vegetation growing season from 2005 to 2023 in the Santun River Irrigation Area in Xinjiang.**

degree of drought was significantly lighter, and the proportions of the area with light drought were 45.04%, 69.43%, and 54.37%, respectively.

**3.3.3 Inter-annual trends of the drought conditions in the study area.** The Sen trend analysis value β of the annual average TVDI in the study area during 2005–2023 was calculated using Equation (4), and Equations (5)–(8) were used to conduct the Mann–Kendall trend significance test. The interannual trend of the TVDI and the spatial distribution of the significance of the superposition of the analysis were obtained using the Python programming code, and based on Table 2, the drought in the Santun River Irrigation Area during the last 20 years. The change trend was divided into six types: extremely significant variable drought, significant variable drought, slight variable drought, slight mitigation, significant mitigation, and extremely significant mitigation. Based on this classification scheme, the spatial distribution of the drought change trend in the Santun River Irrigation Area from 2005 to 2023 was analyzed (Fig 11a). As can be seen from Fig 10a, during the past 20 years, the drought trend in the Santun River Irrigation Area as a whole exhibited a slight drought state, accounting for about 55.32% of the study area. The areas with drought were mostly concentrated in the southeastern part of the Ashley Kazakh Township and the central part of the Liugong Township in the irrigation district. The other areas exhibited a significant drought trend ($P<0.05$), accounting for about 4.21% of the study area. The drought was mostly concentrated in the southeastern part of the irrigation district. The areas characterized by slight mitigation of the regional area ($P<0.05$) accounted for about 34.45%, and these areas were scattered throughout the irrigation area.

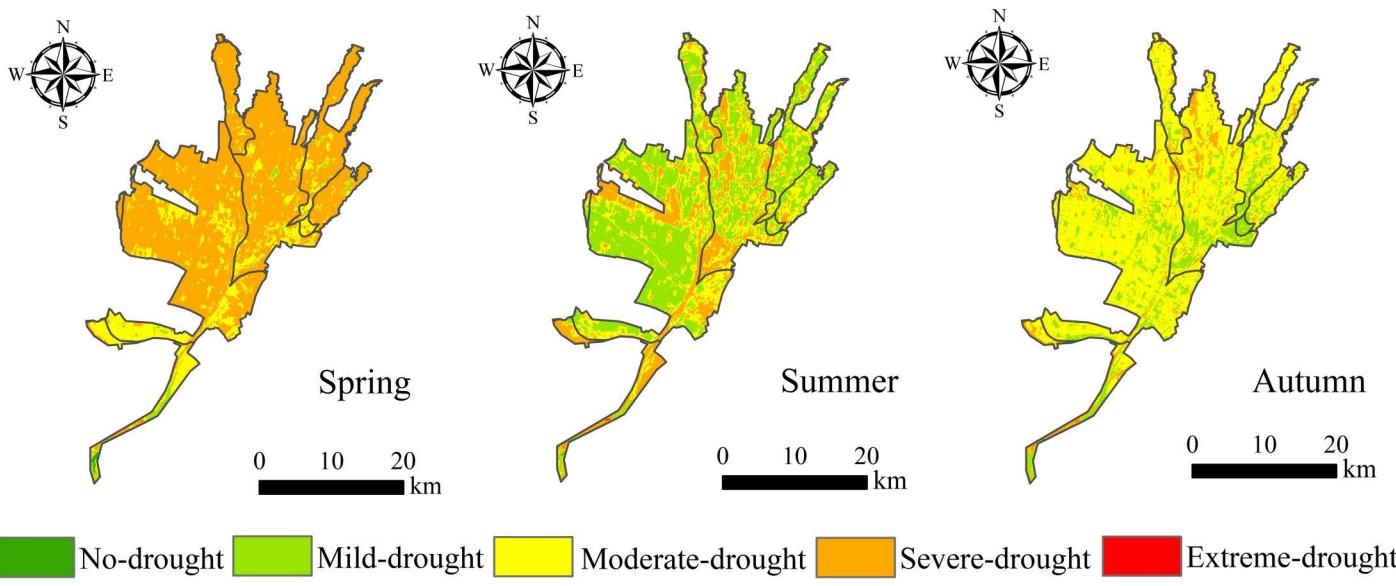

No-drought   Mild-drought   Moderate-drought   Severe-drought   Extreme-drought

**Fig 10. Spatial distributions of the TVDI in the different seasons in the Santun River Irrigation Area, Xinjiang.**

As shown in Fig 11b, the distribution interval of the interannual rate of change of the TVDI in the Santon River Irrigation District from 2005 to 2023 was [–0.016, 0.013]. In addition, in 70.3% of the region, the Sen slope was greater than zero (P < 0.05). This also indicates that most of the Santun River Irrigation Area in Xinjiang was in a state of gradual drought. The drought intensification trend was serious in the southeastern part of the Ashley Kazakh Township and the central part of the Liugong Township, and the drought relief areas were mostly concentrated in the peripheral parts of the irrigation area.

**3.3.4 Multi-year area transfer analysis for different drought classes in the study area.** To further analyze the conversion of the areas with different drought classes during the last 20 years in the Santun River Irrigation Area in Xinjiang, a spatial transfer matrix for the areas with the five drought classes in the irrigation district from 2005 to 2023 was created. As shown in Fig 12, during 2005–2010, in the irrigation area, the area with light drought decreased by 101.66 km², the area with medium drought decreased by 30.46 km², and the area with severe drought increased by 128.61 km². During 2010–2015, the area with light drought decreased by 172.06 km², the area with medium drought increased by 162.41 km2, and the area with severe drought increased by 11.68 km². During 2015–2020, the area with light drought increased by 41.02 km², the area with medium drought increased by 67.89 km², and the area with severe drought increased by 29.02 km². During 2020–2023, the area with light drought increased by 14.25 km², the area with medium drought increased by 13.34 km², and the area with severe drought increased by 30.71 km². Though the areas with extreme drought and no drought fluctuated slightly during the 20-year period, the amplitudes of these changes were very small. During 2005–2010, among the areas with various drought grades that changed to severe drought, the contribution of the change from moderate drought to severe drought was the largest, with an area of 86.03 km² being transferred (54.93%). During 2015–2020, 68.99 km² (72.03%) of the increase in the area with light drought was due to areas with moderate drought becoming areas with light drought. During the past 20 years, the area with light drought in the Santun River Irrigation Area gradually shifted to medium and severe drought, and the area with light drought shifted to medium and severe drought at a rate of 114.9 km² 10 a⁻¹. The areas with medium and severe drought continued to increase, with growth rates of 40.7 km² 10 a⁻¹ and 72.9 km² 10 a⁻¹, respectively. It can be seen that during 2005–2023, in the Santun River Irrigation Area in Xinjiang, the area with light drought gradually shifted to medium and severe drought over time. The proportion of the area with medium drought (72.03%) was the largest during 2015–2020. It can be seen that from 2005

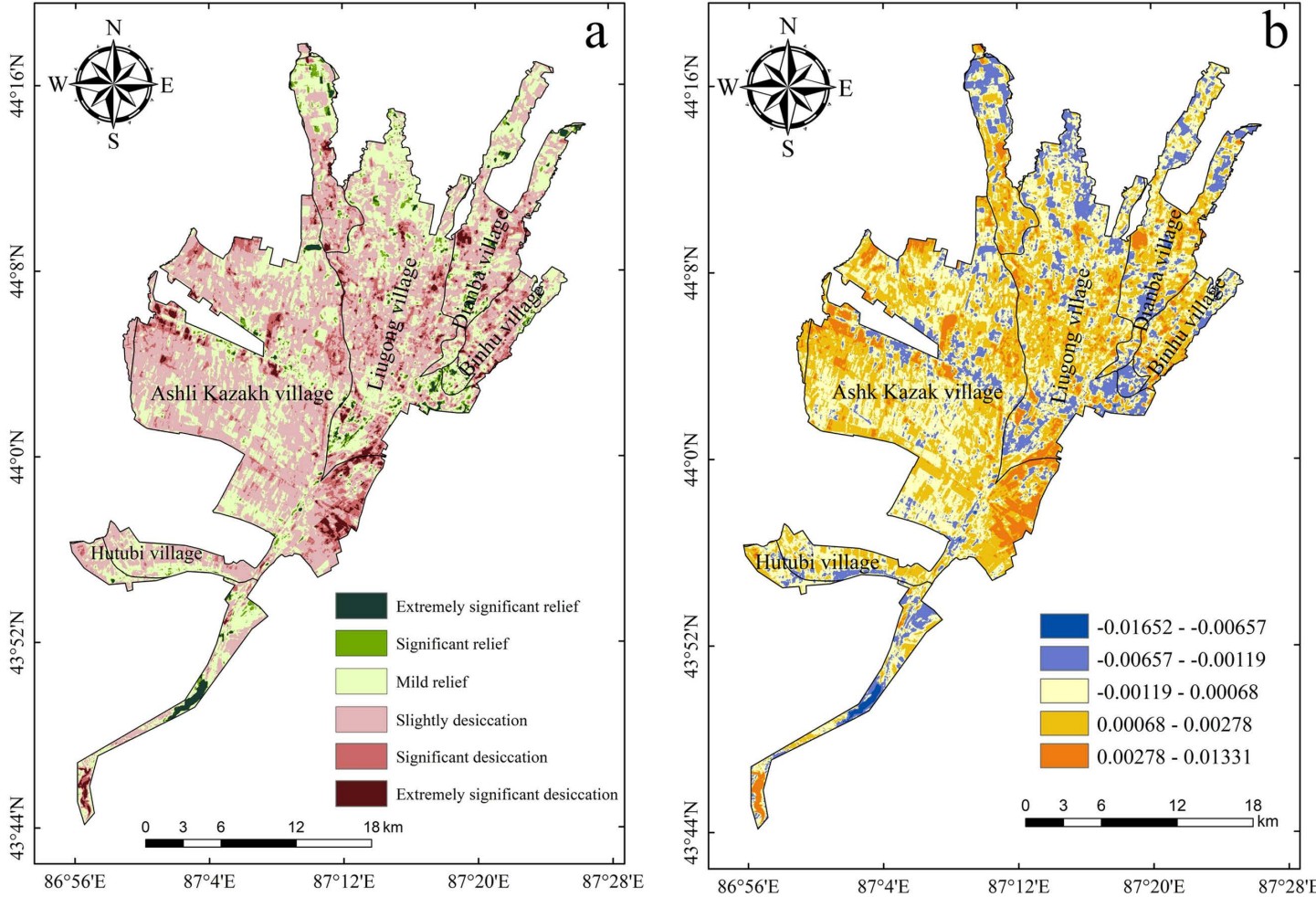

**Fig 11. Types of drought trends in the Santon River Irrigation District and interannual trends of the Sen slope during 2005–2023.** (a) Types of multi-year drought trends; and (b) Inter-annual rate of change of the TVDI.

to 2023, the area with light drought in the Santun River Irrigation Area of Xinjiang gradually shifted to medium and severe drought over time. The areas with no drought and extreme drought were more stable overall, and the changes in their magnitudes were the smallest.

### 3.4  Analysis of drivers of TVDI changes in the Santun River Irrigation Area, Xinjiang

**3.4.1  Detection factor influences and temporal variations.**  With the help of Geodetector, the influences of nine detection factors on the spatial distribution of the TVDI in the Santun River Irrigation Area were calculated using Equation (9). As can be seen from Table 7, the p-values of the nine detection factors are all 0. The q-value of the surface temperature is the largest and passes the 95% significance test, with a q-value of 0.8399, which indicates that temperature was the dominant influence factor affecting the distribution of the TVDI in the Santun River Irrigation Area. The explanatory power of vegetation cover in the study area was greater than 0.4, which indicates that it had a moderate influence on the distribution of the spatial variations in the aridity. The q-value of the land use type was 0.1510, indicating that it influenced the spatial variations in the TVDI in the study area. The q-values of the regional GDP, soil type, and

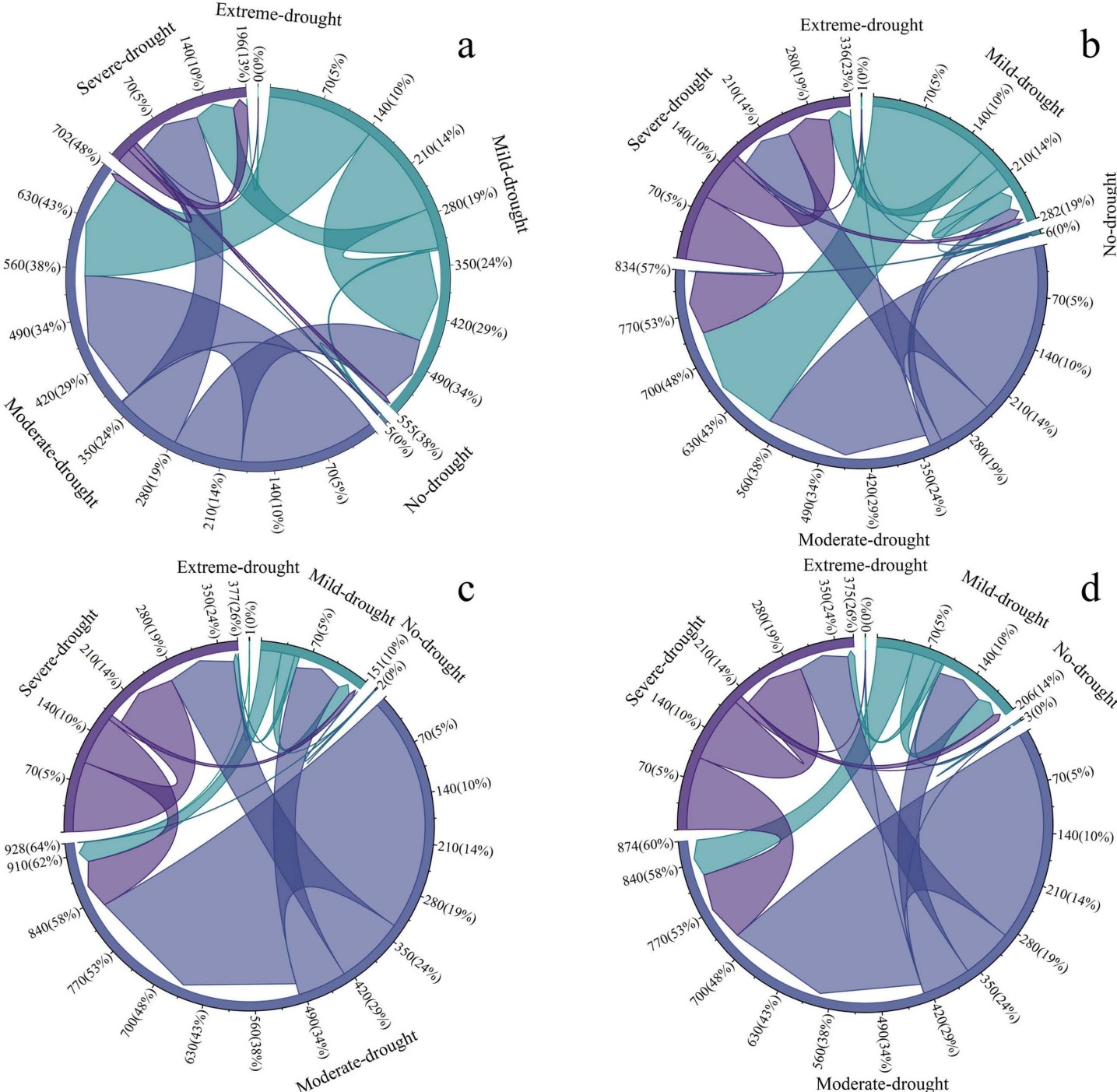

**Fig 12. Area transfers of different drought classes in the Santon River Irrigation District during 2005–2023.** (a) Irrigation district drought transfers during 2005–2010; (b) Irrigation district drought transfers during 2010–2015; (c) Irrigation district drought transfers during 2015–2020; and (d) Irrigation district drought transfers during 2020–2023.

**Table 7. Single-factor detection using Geodetector.**

| Driving factor | q-value | p-value |
|---|---|---|
| Temperature | 0.8399 | 0 |
| NDVI | 0.5310 | 0 |
| FVC | 0.4328 | 0 |
| GDP | 0.0792 | 0 |
| Agrotype | 0.0530 | 0 |
| Aspect | 0.0030 | 0 |
| Slope | 0.0298 | 0 |
| DEM | 0.0539 | 0 |
| Land-use type | 0.1510 | 0 |

altitude elevation were all less than 0.08, so they had less influence on the variations in the TVDI in the Santun River Irrigation Area. Notably, the q-values of the slope and slope direction in the irrigation area were only 0.0298 and 0.003, respectively, indicating that they exerted negligible influences on the variations in the TVDI in the irrigation area.

**3.4.2 Detection of two-way interaction analysis.** To investigate the degrees of influence of the multi-factor interactions on the changes in the drought conditions in the irrigation area, the two-factor interaction detection module of the Geodetector was used to analyze the influences of the two–two interactions among the nine influencing factors on the spatial distribution of the TVDI in the irrigation area (Fig 13). As can be seen from Fig 13, among the nine influencing factors, the effect of the interaction between any two factors on the TVDI significantly exceeded the effect when a single factor acted independently. This effect manifested as nonlinear enhancement or two-factor enhancement, suggesting that the interactions among the nine factors were not simply superimposed and they produced additional effects (Fig 14). As can be seen from Fig 14, none of the nine factors acted independently of the other factors, and the effect of each factor was affected by the other factors, which acted together on the TVDI. Out of all of the interactions between the factors, the interaction that had the greatest influence on the change in the TVDI was the interaction between elevation and temperature, with a two-factor interaction q-value of 0.869. The interaction between the land use type and temperature had the next greatest influence, with a q-value of 0.8568. The double factor interaction was the weakest for the slope and slope direction with a q-value of only 0.044. Considering the geographic location and topography of the irrigation area, it can be concluded that the slope direction did not influence the occurrence of drought to a large extent in the study area due to its flat topography.

## 4. Discussion

### 4.1 Characteristics of temporal and spatial series changes of drought in the Santun River Irrigation Area, Xinjiang

The annual mean TVDI in the Santun River Irrigation Area in Xinjiang was 0.699–0.774 from 2005 to 2023, and it exhibited an overall fluctuating increasing trend. In spring and fall, the drought exhibited an increasing trend, and in summer, the drought eased. The results of the spring and summer studies in the irrigation area are consistent with those of Cheng Jun [45] and Huang Jing et al. [46] on large-scale drought in Xinjiang. The intensification of the spring drought can be attributed to the coupling of natural and human activities. The rise in temperature in spring led to an increase in the vegetation evapotranspiration rate. Coupled with the uneven spatial and temporal distribution of precipitation in the high altitude areas and the market-driven planting structure changes, this aggravated the drought degree under the combined effect of multiple factors [47]. Summer is usually characterized by high temperatures, low precipitation, and high vegetation evapotranspiration, making summer the season of the year that is the most prone to severe drought. However, since the irrigation area mainly relies on glacial meltwater for its water supply and rising temperatures exacerbate glacial

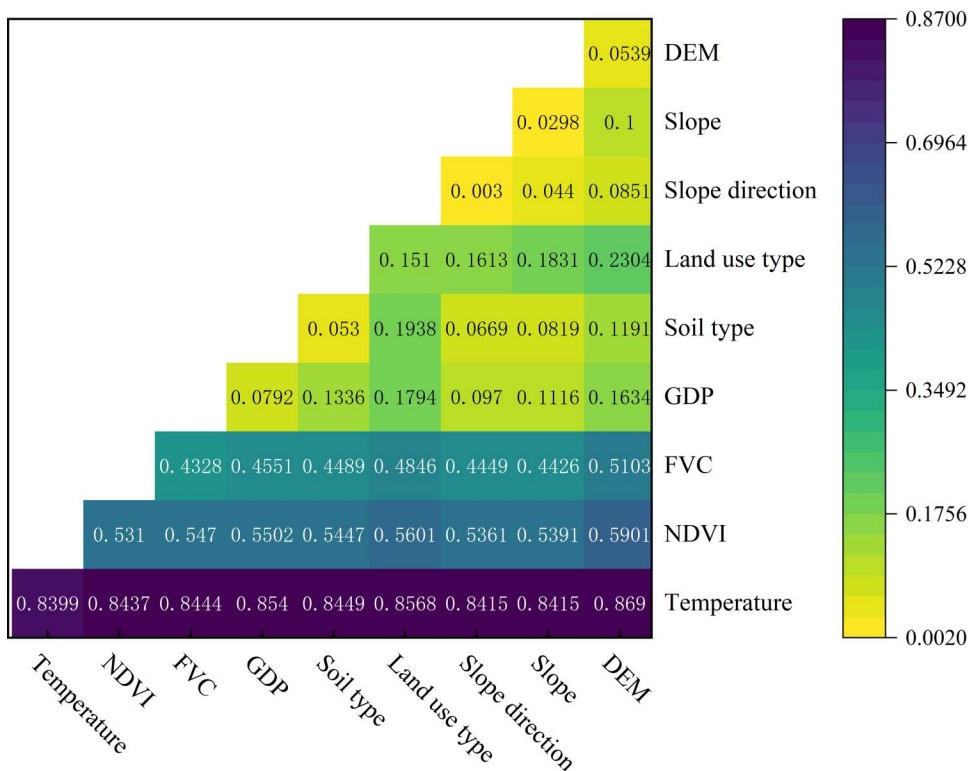

**Fig 13. Detection of drought driver interactions in the Santun River Irrigation Area.**

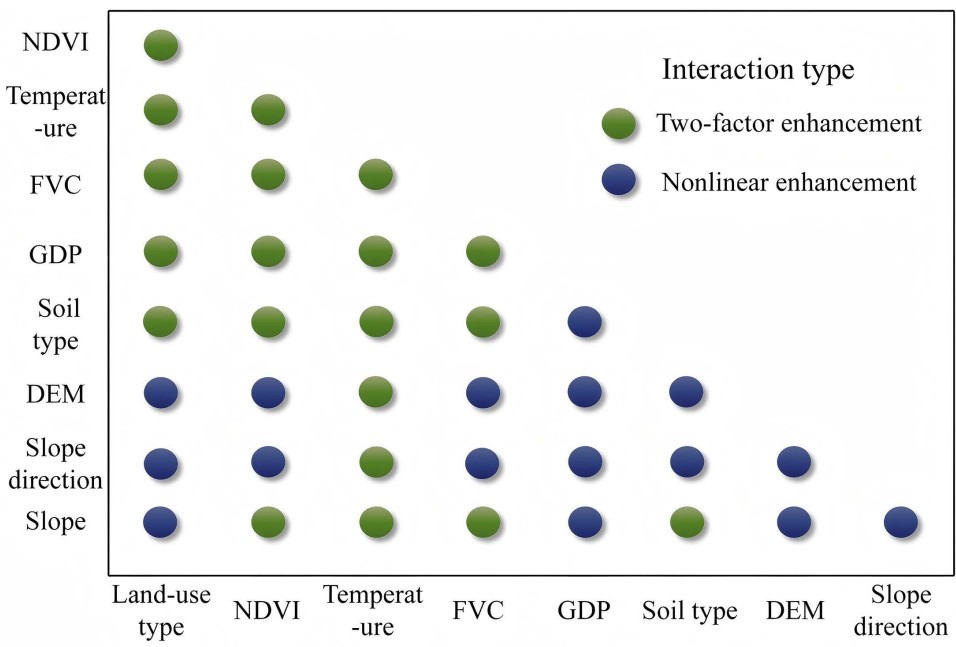

**Fig 14. Types of driver interactions in the Santun River Irrigation Area.**

meltwater, in the rise in temperature increased the surface runoff and raised the groundwater level [48]. In addition, the initial completion of the water diversion project effectively alleviated the water shortage problem in the economic belt on the north slope of the Tianshan Mountains, thus slowing down the occurrence of drought. Regarding the increase in the autumn drought degree, there is some difference between the research results and the reduction in the autumn drought degree in northern Xinjiang reported by Cheng Jun et al. [45]. The reason for this may be that with decreasing temperature, the evapotranspiration of vegetation water decreases and the large-scale regional drought eases. However, in the irrigated area, the crops are in the sowing period, the water demand of the vegetation is large, and the regional precipitation is scarce and its spatial and temporal distribution is uneven. This situation is more likely to lead to insufficient soil moisture, thus aggravating the occurrence of autumn drought [49].

The spatial distribution of the drought in the irrigated area exhibited obvious spatial heterogeneity, i.e., the drought was higher in the northern region than that in the southern region. The reason for this spatial distribution may be that the Santun River Irrigation Area is located on the northern slope of Tianshan Mountains in Xinjiang, and water vapor transport is blocked by the mountains, so the main source of the water resources is glacier melt water. The southern part of the irrigation area is closer to the Tianshan Mountains, and the water resources are more abundant compared to the northern part. The northern part of the irrigation area is closer to the Gulban Desert, and the vegetation coverage is more sparse than in the southern part, with a weak soil and water conservation ability and low irrigation efficiency. In addition, the northern plain has a high level of urbanization, and groundwater over-extraction is serious, which is more likely to induce drought [50]. Moreover, the implementation effect of irrigation policies in the irrigation areas is uneven, and the distribution of the water resources is unequal, which further intensifies the spatial heterogeneity of the drought.

It was found that more than 60% of the study area exhibited a trend of continuous drought, and over the past 20 years, the areas with a mild drought grade shifted to medium and severe drought grades at a rate of 114.9 $km^2 \cdot 10\ a^{-1}$. In addition to natural factors, human activities also play an important role in promoting this phenomenon [51]. Taking the land use data for 2005 and 2020 as an example, in this study, the land use type area transfer matrix from 2005 to 2020 was obtained (Table 8). The results show that compared with 2005, the area of grassland in 2020 was 18.199 $km^2$ smaller, while the area of the urban and rural building land was 44.698 $km^2$ greater, reflecting the rapid transformation of land into urban and rural building land in the process of urbanization. In addition, the original forest area in the irrigated area was redistributed, and up to 80.17% of the area was converted to cultivated land area, which further reflects the cultivated land expansion trend of the land use change pattern. Urban expansion will lead to the over-exploitation of groundwater near irrigated areas in order to meet living and irrigation needs, which will further lead to the gradual loss of the water storage capacity of the underground aquifers, causing a chain reaction that increases the likelihood of drought [52]. In addition, urban expansion has increased greenhouse gas emissions and created the conditions for drought [53].

**Table 8. Area transfer matrix for land use types in the Santun River Irrigation Area during 2005–2020.**

| Area (km²) | | 2020 | | | | | |
|---|---|---|---|---|---|---|---|
| | | Grassland | Farmland | Urban and rural residential land | Water bodies | Unutilized land | Transferred out |
| 2005 | Grassland | 48.352 | 44.651 | 18.235 | 2.822 | 0.066 | 114.126 |
| | Farmland | 40.186 | 465.298 | 36.171 | 2.071 | 0.000 | 543.725 |
| | Urban and rural residential land | 2.446 | 12.564 | 13.345 | 0.184 | 0.000 | 28.539 |
| | Woodland | 1.030 | 5.985 | 0.450 | 0.000 | 0.000 | 7.465 |
| | Water bodies | 0.535 | 8.709 | 4.228 | 10.435 | 1.536 | 25.443 |
| | Unutilized land | 3.378 | 3.436 | 0.807 | 0.020 | 0.000 | 7.641 |
| | transferred in | 95.927 | 540.642 | 73.237 | 15.532 | 1.602 | / |

## 4.2 Drivers of drought in the Santun River Irrigation Area, Xinjiang

The results of this study show that temperature had the greatest effect on drought (q = 0.8399) under the one-way test. It has been widely proven that temperature is one of the main factors causing drought [54]. Considering the geographic location and climatic conditions of the Santun River Irrigation area, the above results may be due to the fact that the Santun River Irrigation area is located in the interior of the Eurasian continent, where the high temperatures and small amount of precipitation lead to the occurrence of drought being mainly influenced by temperature. In addition, the water resources in the study area mainly come from melting of snow in the Tianshan Mountains, and the generation of melt water from glaciers in the Tianshan Mountains is greatly promoted when the temperature rises, while the warm and wet climate is extremely conducive to plant growth and development and alleviates regional drought [55]. Therefore, the q-values of influencing factors such as the temperature and NDVI are generally large.

Regarding the two-factor interactions, the interaction between elevation and temperature had the highest explanatory power on drought (q = 0.869). Several studies have confirmed that a combination of factors such as temperature and elevation can indeed affect vegetation dynamics, resulting in drought [56–58]. The reason for this may be the dual regulation of the water cycle process by the altitude-temperature coupling. First, the change in elevation directly affects the distributions of the temperature and precipitation. Higher elevations have lower temperatures, less evaporation, and more precipitation. At lower elevations, the temperature is higher, the evaporation is large, and precipitation is scarce, making soil water loss easier [59]. In addition, elevation differences also affect the path and speed of surface runoff. During the transport of precipitation and snowmelt runoff from high altitude areas to low altitude areas, part of the water is evaporated and absorbed by vegetation, which further aggravates the degree of drought in the low altitude areas [60]. This interaction between temperature and elevation affects the spatial distribution of drought by influencing the water cycle and the evaporation and precipitation distributions in the study area.

## 4.3 Drought coping strategies

The drought characteristics in the Santun River irrigation area in Xinjiang are significant, and they have a substantial influence on the regional water resources and agricultural production planning. Therefore, future water resource management needs to focus on improving the water use efficiency and optimizing water resource allocation. For example, strategies should include promoting the development of information technology, establishing a sound drought early warning and forecast system, improving the monitoring efficiency, developing suitable crop cultivation patterns according to the terrain and climate differences, and considering the local water resource allocation to appropriately increase the external water transfer. Additional measures that can be taken include actively adjusting the planting structure of crops in irrigated areas, prioritizing drought-tolerant and heat-resistant crop varieties in northern arid areas, introducing low-water consumption crops according to national policies to improve crop adaptability to drought, continuing to promote water-saving irrigation technology, reducing the ineffective evaporation and leakage of water resources, improving the utilization efficiency of irrigation water, increasing the input of soil moisture monitoring instruments and the real-time monitoring of soil moisture in irrigated areas, and formulating reasonable irrigation plans according to the soil moisture content and crop water requirements.

The implementation of these comprehensive measures can effectively reduce the negative impact of drought in irrigated areas while ensuring the stability of agricultural production and the safety of peoples' lives.

## 4.4 Research limitations and prospects

In this study, although the inversion of TVDI drought monitoring in irrigated areas using Landsat series remote sensing data achieved good results overall, it should be noted that cloud cover and sensor accuracy affect the inversion of the

 

index. The TVDI is a combination of the LST and vegetation index (NDVI) and is used to assess drought conditions, which may differ in areas with different climate conditions and vegetation types. The TVDI is more suitable for areas with good vegetation coverage, while in areas with bare soil or sparse vegetation, the vegetation coverage is low and the NDVI value is low or even close to zero, and thus, it cannot accurately reflect the real water status of the surface. In addition, in bare soil areas, the surface temperature is affected by many factors (soil type, solar radiation, and wind speed), leading to an incomplete correlation between the change in the LST and the soil moisture. Thus, it is difficult to build a stable feature space, which ultimately leads to a poor monitoring effect for the TVDI. Moreover, different types of soil have different thermal inertia, which may lead to differences in the response of the surface temperature to the soil moisture. This difference may be more pronounced in areas with exposed soil, affecting the applicability of the TVDI index. In view of these limitations, the utilization of comprehensive assessment, combined with other drought indicators (such as the soil moisture index and rainfall index,), will be more accurate in future studies. When the study area is focused on an irrigation area, the spring crops are in the greening stage, there is more bare land, and the vegetation coverage is low. The inversion of the TVDI value using remote sensing data will correspondingly improve, and the accuracy of the drought assessment may decrease. However, summer and autumn are the peak seasons of vegetation growth, so the vegetation coverage is better, and the inversion of the TVDI value using remote sensing data is more accurate.

Although the temporal and spatial driving mechanism of drought in the Santun River irrigated area has been systematically analyzed, the influence of groundwater depletion on regional drought, which is one of the important reasons for drought in arid irrigated area, has not been quantified. In addition, the planting structure, crop types, irrigation methods, and government policies affect drought in irrigated areas. Therefore, in order to accurately assess the drought conditions in irrigated areas, a single index may not be comprehensive enough, and it is necessary to integrate multi-source data for comprehensive analysis, which will become an important direction of drought research in irrigated areas in the future. Although the Landsat series satellite data used in this study are suitable for regional analysis, the monitoring accuracy of the local drought heterogeneity in irrigated areas is not as high. In follow-up studies, the temporal and spatial resolutions of the data should be improved, and the drought research should be refined to capture the key short-term drought fluctuations during the crop growth period and improve the sensitivity and identification ability of the local heterogeneity in small-scale irrigation areas. In addition, since there is no clear document stipulating the drought grade classification criteria, the classification criteria for drought in different regions are different, which leads to differences in the conclusions of different studies.

## 5. Conclusions

In this study, the TVDI was used to quantify the degree of drought in the Santun River Irrigation Area, Xinjiang, and the spatiotemporal characteristics of the evolution, change trend, and grade transfer of drought in the region from 2005–2023 were investigated. In addition, the driving factors affecting drought were analyzed based on the geographic detector model. The main conclusions of this study are summarized below.

1. Temporally, the TVDI in Santun River Irrigation Area in Xinjiang was 0.699–0.774 and increased slowly at a rate of 0.008 $a^{-1}$, and the drought exhibited an aggravating trend. Spatially, the spatial distribution of the TVDI in Santun River Irrigation Area in Xinjiang exhibited significant heterogeneity; that is, the drought was higher in the northern region than in the southern region.

2. From 2005 to 2023, the rate of change in the TVDI in the Santun River Irrigation Area in Xinjiang was between −0.016 and 0.013, and the Sen slope was greater than zero in more than 61.87% of the region. In the past 20 years, the areas with a light drought grade in the study area shifted to moderate and severe drought grades at a rate of 114.9 $km^2 \cdot 10 \, a^{-1}$.

3. The spatial heterogeneity of the TVDI in Santun River Irrigation Area in Xinjiang was influenced by many factors. Based on the single-factor analysis, temperature was the main factor influencing the drought, with a q-value of > 0.83. Regarding the two-factor interactions, the interaction between temperature and elevation dominated the spatial differentiation of the drought (q-vale of 0.869).

To effectively deal with the challenges of drought in the future, it is important to improve monitoring accuracy, integrate multi-source data for comprehensive analysis, and quantify the impact of groundwater depletion on drought in follow-up research. In addition, it is necessary to determine suitable planting patterns and adjust water resources management policies according to the different regional ecological environments. These measures can effectively reduce the negative effects of drought in irrigated areas, ensuring the stability of agricultural production and the safety of peoples' lives in irrigated areas.

## Supporting information

**S1 File. The original data file of the manuscript graphics.**
(XLSX)

## Author contributions

**Conceptualization:** Yan Xu.

**Data curation:** Chunlei Lu.

**Writing – original draft:** Yuxin Wei.

**Writing – review & editing:** Hongfei Tao, Mahemujiang Aihemaiti, Youwei Jiang, Qiao Li.

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
