## [Decision Letter · Decision Letter 0]

13 Dec 2024

PONE-D-24-49768Analysis of growing season drought characteristics and driving factors for vegetation in the Santun River Irrigation Area in XinjiangPLOS ONE

Dear Dr. Li,

Thank you for submitting your manuscript to PLOS ONE. After careful consideration, we feel that it has merit but does not fully meet PLOS ONE’s publication criteria as it currently stands. Therefore, we invite you to submit a revised version of the manuscript that addresses the points raised during the review process.

We look forward to receiving your revised manuscript.

Kind regards,

Nguyen-Thanh Son, Ph.D.

Academic Editor

PLOS ONE

Journal Requirements:

This research was funded by Major Project of Xinjiang Uygur Autonomous Region (2023A02002-1),National Natural Science Foundation of China(41762018), Open Project of Xinjiang Key Laboratory of Water Conservancy Engineering Safety and Water Disaster Prevention (ZDSYS-JS-2021-09), 2023 Research project of Xinjiang Key Laboratory of Water Conservancy Engineering Safety and Water Disaster Prevention (ZDSYS-YJS-2023-10) and The Belt and Road Special Foundation of the National Key Laboratory of Water Disaster Prevention (2020491611). 

4. We note that Figures 1, 8, 9, and 10 in your submission contain [map/satellite] images which may be copyrighted. All PLOS content is published under the Creative Commons Attribution License (CC BY 4.0), which means that the manuscript, images, and Supporting Information files will be freely available online, and any third party is permitted to access, download, copy, distribute, and use these materials in any way, even commercially, with proper attribution. For these reasons, we cannot publish previously copyrighted maps or satellite images created using proprietary data, such as Google software (Google Maps, Street View, and Earth). For more information, see our copyright guidelines: http://journals.plos.org/plosone/s/licenses-and-copyright.

a. You may seek permission from the original copyright holder of Figures 1, 8, 9, and 10 to publish the content specifically under the CC BY 4.0 license.  

Reviewers' comments:

Reviewer's Responses to Questions

**Comments to the Author**

1. Is the manuscript technically sound, and do the data support the conclusions?

Reviewer #1: Yes

Reviewer #2: Yes

Reviewer #3: Partly

2. Has the statistical analysis been performed appropriately and rigorously? 

Reviewer #1: Yes

Reviewer #2: Yes

Reviewer #3: Yes

3. Have the authors made all data underlying the findings in their manuscript fully available?

Reviewer #1: No

Reviewer #2: Yes

Reviewer #3: No

4. Is the manuscript presented in an intelligible fashion and written in standard English?

Reviewer #1: Yes

Reviewer #2: Yes

Reviewer #3: Yes

5. Review Comments to the Author

Reviewer #1: 1. The introduction of the manuscript emphasizes the insufficiency of current research in integrating human activities with drought issues. However, the discussion on the impacts of land use changes and human activities on drought remains superficial, lacking an in-depth exploration of the underlying driving mechanisms. Furthermore, the study fails to adequately address the core questions raised in the introduction.

2. While the paper employs the TVDI indicator, which is suitable for areas with substantial vegetation cover, its applicability in regions with sparse vegetation or bare soil is debatable. The manuscript does not sufficiently address the limitations of this indicator or its potential impact on the results. A more thorough discussion of these limitations would enhance the credibility of the findings.

3. The study primarily relies on annual temporal resolution. Although seasonal variations are mentioned, the analysis does not capture critical short-term drought fluctuations within key growth periods, which diminishes its direct applicability to agricultural production and water resource management. Additionally, the use of a 30-meter spatial resolution, though appropriate for large-scale regional analysis, may obscure local heterogeneity in small-scale irrigation areas, reducing its utility for precise monitoring and strategy development.

4. Although the manuscript employs the GeoDetector model to reveal the independent and interactive effects of various driving factors, the explanations of these interactions remain qualitative, lacking robust scientific reasoning. Specifically, the paper identifies the interaction between elevation and temperature as having the strongest explanatory power for drought distribution but fails to elucidate the underlying physical mechanisms behind this phenomenon.

5.The introduction highlights the large-scale impacts of drought and critiques the limitations of large-scale studies in integrating human activities and regional heterogeneity. However, the research itself is focused on the Santun River Irrigation Area, a small-scale region, analyzing local drought characteristics and drivers. This shift between the research objectives and study scope lacks clear logical coherence, resulting in a degree of inconsistency in the manuscript.

6.The correlation analysis between TVDI and soil moisture does not report p-values to verify the strength of the relationship. The GeoDetector model results, including single-factor (e.g., temperature, NDVI) and interaction effects, lack significance tests to confirm the robustness of the q-values. The Mann-Kendall trend analysis for seasonal and annual TVDI variations omits the significance levels (e.g., p-values) of the identified trends.

7.In the discussion of seasonal drought variations, the reasons for the intensification of spring drought and the alleviation of summer drought are not analyzed in depth. The explanation is limited to temperature changes or irrigation water use, without considering factors such as precipitation distribution or adjustments in crop planting structures. Additionally, for the spatial heterogeneity where drought is more severe in the northern region, it is merely described as being "close to the desert with sparse vegetation," without a detailed analysis of human activities (e.g., over-extraction of water resources, urban expansion) or natural factors (e.g., soil water retention capacity).

Reviewer #2: 1. What is the difference between the canopy temperature in VSWI and LST?

2. Are there any other similar studies examining the small scales other than the study area mentioned in the paper?

3. There are a number of abbreviations in the text, and it is recommended that a table or an appendix be added listing all abbreviations and their full names.

4.

Line 130: amax and bmax should use suffix notation.

Line 131: amin and bmin should use suffix notation.

Line 281: 2020 should be 2023

5. Reference 34 and 42 are the same

Reviewer #3: This paper focuses on the analysis of growing season drought characteristics and their driving factors for vegetation in the Santun River Irrigation Area in Xinjiang, a topic of significant scientific and practical value. However, there are deficiencies in data analysis and result discussion that require substantial revision and supplementation.

1. Data sources are not clearly specified. There is a lack of descriptions regarding the time range and spatial resolution of the data sources in Table 2. The authors mention choosing 60 images from 2005 to 2023, but the frequency of the dataset seems low. The authors need to justify whether these calculation results can represent quarterly drought characteristics.

2. Similarly, the soil moisture data in Table 2 need to include information about its spatial extent. If these data are discrete, the interpolation methods also need to be described.

3. The paper focuses on vegetation area, but some areas without vegetation are also included in the statistics, such as urban-rural residential land, water bodies, and unused land. Excluding these areas might yield more accurate results.

4. In section 3.3.1, the paper statistically analyzes the annual average TVDI and the percentages of the area with different drought classes from 2005 to 2023. The methodology and its rationale should be explained; additionally, the paper should clarify how the calculated drought areas were validated.

5. The explanation for Fig 11 is unclear, with no introduction of the numbers before the area percentages. Additionally, Fig 11 (a) and Fig 11 (b) are both labeled "Irrigation area transfers," while Fig 11 (c) and Fig 11 (d) are labeled "Irrigation district drought transfers." The descriptions should be consistent.

6. Section 3.4 analyzes the influence of driving factors but fails to explain their rationale and calculation methods. Moreover, precipitation, discussed as an influential factor in sections 4.1 and 4.2, is not mentioned among the driving factors.

7. Some discussions in this manuscript are inadequate. The inference of drought driving factors in section 4.3 and the discussion of drought response strategies in section 4.4 should focus on the drought driving factors studied in this paper. The current arguments seem disconnected from the research content and should be revised to align with the research findings.

8. Some reference formats are inconsistent with the journal’s requirements and should be checked and revised.

In conclusion, this paper has significant deficiencies in data introduction, research method description, data analysis, and result discussion. Major revision is recommended.

6. PLOS authors have the option to publish the peer review history of their article (what does this mean? ). If published, this will include your full peer review and any attached files.

**Do you want your identity to be public for this peer review?** For information about this choice, including consent withdrawal, please see our Privacy Policy .

Reviewer #1: No

Reviewer #2: No

Reviewer #3: No

---

## [Author Response · Author response to Decision Letter 1]

25 Dec 2024

Response to Reviewers

Thank you for your letter and thank the reviewers and editorial department for their comments on our manuscript. We have revised the manuscript with reference to the reviewer's comments for your approval. Our reply is as follows:

Reviewer #1

1.The introduction of the manuscript emphasizes the insufficiency of current research in integrating human activities with drought issues. However, the discussion on the impacts of land use changes and human activities on drought remains superficial, lacking an in-depth exploration of the underlying driving mechanisms. Furthermore, the study fails to adequately address the core questions raised in the introduction.

Answer: Based on your suggestions, we have reorganized the discussion section of the paper to focus on an in-depth discussion of the impacts of human activities and land-use change types on drought and to explore in depth the underlying driving mechanisms. Please refer to lines 4879-484,495-522.

Regarding the issue of introduction, after an in-depth discussion by the team, we have reorganized the introduction and core issues of this paper.The core of this paper is to calculate the temperature-vegetation drought index (TVDI) of the study area, and to clarify the spatial and temporal distribution characteristics of drought and the influencing factors of drought in the Santun River Irrigation Area of Xinjiang during the period of 2005-2023 by combining the methods of trend analysis, geo-probe and spatial transfer matrix, so as to provide a scientific basis for optimizing the allocation of water resources, adjusting the irrigation system, and guaranteeing the stability of agricultural production.Therefore, we have revised the introductory section of this paper to ensure logical consistency in the research objectives and scope of the study.Please refer to lines 72-96.

2.While the paper employs the TVDI indicator, which is suitable for areas with substantial vegetation cover, its applicability in regions with sparse vegetation or bare soil is debatable. The manuscript does not sufficiently address the limitations of this indicator or its potential impact on the results. A more thorough discussion of these limitations would enhance the credibility of the findings.

Answer: Based on your suggestions, we have reorganized the discussion section to focus on adding a discussion of the potential impact of the limitations of the TVDI metrics on the results.please refer to lines 587-604.

3.The study primarily relies on annual temporal resolution. Although seasonal variations are mentioned, the analysis does not capture critical short-term drought fluctuations within key growth periods, which diminishes its direct applicability to agricultural production and water resource management. Additionally, the use of a 30-meter spatial resolution, though appropriate for large-scale regional analysis, may obscure local heterogeneity in small-scale irrigation areas, reducing its utility for precise monitoring and strategy development.

Answer: Thank you for your valuable comments. We quite agree with you. At the same time, I also recognize the shortcomings in the research, and sincerely thank you for your correction. However, since the current research for drought characteristics and influencing factors in the Santun River Irrigation Area of Xinjiang is still in a blank state, the main purpose of this paper is to fill the research gaps in the field of drought in the region, to provide basic data and preliminary understanding for the subsequent related research, and to pave the way for the later refinement of the study in the irrigation district.Therefore, we chose the Landsat series data, which is widely applicable, has a long time series, is easy to obtain, and has more comprehensive information on spectral bands, to analyze the long time-series drought characteristics and influencing factors of the Santun River irrigation area in Xinjiang. In response to the issues you mentioned, we plan to further improve the temporal and spatial resolution of the data (Sentinel-2, ground-based observations) in future studies, and refine the analysis of droughts in the region by combining with the type of cultivation in the irrigation area and the irrigation system, etc., in order to capture critical short-term drought fluctuations during the crop growth period and to improve the sensitivity to and ability to identify localized heterogeneity in small-scale irrigation areas.Following your suggestion, we have added a discussion of this deficiency in Chapter 4.Please refer to lines 612-617.

4.Although the manuscript employs the GeoDetector model to reveal the independent and interactive effects of various driving factors, the explanations of these interactions remain qualitative, lacking robust scientific reasoning. Specifically, the paper identifies the interaction between elevation and temperature as having the strongest explanatory power for drought distribution but fails to elucidate the underlying physical mechanisms behind this phenomenon.

Answer: Following your suggestion, we have reorganized and deepened the driver discussion section of this article, adding a discussion on the underlying physical mechanisms behind this phenomenon.Please refer to lines 546-562.

5.The introduction highlights the large-scale impacts of drought and critiques the limitations of large-scale studies in integrating human activities and regional heterogeneity. However, the research itself is focused on the Santun River Irrigation Area, a small-scale region, analyzing local drought characteristics and drivers. This shift between the research objectives and study scope lacks clear logical coherence, resulting in a degree of inconsistency in the manuscript.

Answer: Thank you for your valuable comments. We have revised the introductory section of this paper to ensure logical consistency in the research objectives and scope of the study.Please refer to lines 72-96.

6.The correlation analysis between TVDI and soil moisture does not report p-values to verify the strength of the relationship. The GeoDetector model results, including single-factor (e.g., temperature, NDVI) and interaction effects, lack significance tests to confirm the robustness of the q-values. The Mann-Kendall trend analysis for seasonal and annual TVDI variations omits the significance levels (e.g., p-values) of the identified trends.

Answer: According to your suggestion, we have added p value to the correlation analysis between TVDI and soil moisture in the text,please refer to lines 214-215; In the single-factor analysis of geographic detector, p-value was added, and the original q-value map was replaced with a single-factor analysis table,please refer to lines 401,412; Reorganize the data and add the significance level in the trend analysis,please refer to lines 344-350.

7.In the discussion of seasonal drought variations, the reasons for the intensification of spring drought and the alleviation of summer drought are not analyzed in depth. The explanation is limited to temperature changes or irrigation water use, without considering factors such as precipitation distribution or adjustments in crop planting structures. Additionally, for the spatial heterogeneity where drought is more severe in the northern region, it is merely described as being "close to the desert with sparse vegetation," without a detailed analysis of human activities (e.g., over-extraction of water resources, urban expansion) or natural factors (e.g., soil water retention capacity).

Answer: According to your suggestion, we reorganized the discussion section to deepen the discussion on the causes of the seasonal drought. At the same time, the discussion of arid spatial heterogeneity phenomenon adds to human activities, natural factors and other contents. Make the content more substantial and persuasive.Please refer to lines479-484,495-522.

Reviewer #2

1.What is the difference between the canopy temperature in VSWI and LST?

Answer Thank you for your valuable comments. The main parameter in this study is TVDI, and the one used is based on the surface temperature (LST) obtained by thermal infrared sensors carried by the Landsat series of satellites.

Calculating VSWI uses canopy temperature, which is the temperature of plants and/or vegetation at the surface and is commonly used to indicate the heat and moisture status of surface vegetation. Whereas, the calculation of TVDI uses the Land Surface Temperature (LST), which is an important parameter for heat exchange at the interface between the atmosphere and the land surface, and reflects the degree of heating and cooling of surface objects.The main difference between the two of them is:

1)The measurement object is different: the canopy temperature mainly measures the temperature of the crop canopy (such as stems and leaf surfaces), which more directly reflects the moisture and heat status of the crop itself. Surface temperature, on the other hand, measures the temperature of the entire surface, including soil, vegetation, water bodies, etc. In remote sensing applications, it is usually measured as the temperature of the entire surface, including soil, vegetation, water bodies, etc. In remote sensing applications, it usually represents the average or composite temperature over a wider area.

2)Different influencing factors: Canopy temperature is influenced by a variety of factors, including soil moisture status, crop transpiration, and environmental factors (e.g., wind speed, temperature, and solar radiation). Surface temperatures, on the other hand, are more influenced by solar radiation, type of ground cover, atmospheric conditions, and other factors

2. Are there any other similar studies examining the small scales other than the study area mentioned in the paper?

Answer Based on a comprehensive review of the relevant literature, I observed that relevant studies conducted for small scales such as irrigated areas exist. Specific relevant studies are listed below:

1)Yue Qiong and other scholars, utilized the CFSMP model to analyze the seasonal drought in irrigation areas in South China. The results showed that the CSFMP model has the potential to mitigate seasonal drought in South China and can be applied to similar regions with comparable resource crises.(Reference Qiong Y ,Ningbo C ,Fan Z , et al. Adaptation to seasonal drought in irrigation districts of south China: A copula-based fuzzy-flexible stochastic multi-objective approach for precise irrigation planning [J]. Journal of Hydrology, 2023, 625 (PA):)

2)Zhang Fan and other scholars, proposed a Copula-based stochastic multi-objective planning (C-SMP) model for optimizing irrigation strategies to mitigate the negative impacts of seasonal agricultural drought on the yields of different crops in the irrigation area of Dongfeng Reservoir, Meishan City, Southwest China.(Reference Fan Z ,Ningbo C ,Shanshan G , et al. Irrigation strategy optimization in irrigation districts with seasonal agricultural drought in southwest China: A copula-based stochastic multiobjective approach [J]. Agricultural Water Management, 2023, 282)

3)Akinwale T et al. Spatio-temporal characterization of drought in northern Nigeria using the self-calibrated Palmer Drought Severity Index (sc-PDSI).(Reference Ogunrinde T A ,Oguntunde G P ,Olasehinde A D , et al. Drought spatiotemporal characterization using self-calibrating Palmer Drought Severity Index in the northern region of Nigeria [J]. Results in Engineering, 2020, 5 (C): 100088-100088.)

In summary, research exists to study drought characteristics at small scales or within specific regions. Our study clarifies the characteristics and influencing factors of drought in the Santun River Irrigation Area of Xinjiang through trend analysis, geoprobes and spatial transfer matrices. This is crucial for optimizing water resource allocation in the irrigation area, ensuring stable agricultural production and promoting sustainable agricultural development.

3.There are a number of abbreviations in the text, and it is recommended that a table or an appendix be added listing all abbreviations and their full names.

Answer Thank you for your comments.Full name and abbreviation control table has been added.Please refer to line 129.

4.Line 130: amax and bmax should use suffix notation.

Answer This has been modified; please refer to line 149.

5.Line 131: amin and bmin should use suffix notation.

Answer This has been modified; please refer to lines 149.

6.Line 281: 2020 should be 2023

Answer This has been modified; please refer to line 326.

7.Reference 34 and 42 are the same

Answer Thank you for your comments.This has been modified.

Reviewer #3:

1.Data sources are not clearly specified. There is a lack of descriptions regarding the time range and spatial resolution of the data sources in Table 2. The authors mention choosing 60 images from 2005 to 2023, but the frequency of the dataset seems low. The authors need to justify whether these calculation results can represent quarterly drought characteristics.

Answer Thank you for your comments.In response to your suggestion,we have modified Table 1 to add data time ranges.Please refer to line 127; The selection of the dataset was based on strict selection criteria after excluding effects such as cloudiness and weather, and we ensured that the selected images were evenly distributed in time and covered key periods in each season to ensure that they accurately reflect the drought conditions in each season. Meanwhile, we compared the results of remote sensing calculations with the soil water content observed on the ground, and found that there was a good correlation between the two, indicating that our remote sensing calculations could better reflect the drought conditions on the ground. Compared with similar studies, we believe that the datasets used in this study are already more accurate.

2.Similarly, the soil moisture data in Table 2 need to include information about its spatial extent. If these data are discrete, the interpolation methods also need to be described.

Answer According to your recommendations..The measured soil moisture content data in this paper are continuous data.We have added information on the spatial extent of the sampling points for measured soil moisture content in the irrigation area, as detailed in Figure 3.Please refer to lines 218-219.

3.The paper focuses on vegetation area, but some areas without vegetation are also included in the statistics, such as urban-rural residential land, water bodies, and unused land. Excluding these areas might yield more accurate results.

Answer Thank you for your valuable comments. When analyzing in this study, we mainly consider that non-vegetated areas are equally important for the study of the functioning of the overall ecosystem and the effects of drought, as they may indirectly affect the growth and distribution of vegetation. For example, urban and rural residential land use may indirectly affect the surrounding vegetation by changing the surface cover, affecting the hydrological cycle, and so on. Therefore, we have discussed and inferred about it in the discussion section. Please refer to lines 495-522. In the follow-up study, we will use more precise geographic information technology tools to identify and extract vegetation areas to ensure the accuracy of the study. Thank you again for your review and guidance!

4.In section 3.3.1, the paper statistically analyzes the annual average TVDI and the percentages of the area with different drought classes from 2005 to 2023. The methodology and its rationale should be explained; additionally, the paper should clarify how the calculated drought areas were validated.

Answer Thank you for your comments.In this study, the method used to analyze the trend of annual average TVDI change is linear trend analysis.The analysis of the percentage of different drought types is mainly done by using ArcGIS software to extract the data in the raster image, followed by statistical analysis in Excel, and after integrating the data, the plotting of the percentage of stacked area is carried out with the help of Origin software.Based on your suggestion, we are adding instructions on how to use it, in our analysis.Please refer to line

---

## [Decision Letter · Decision Letter 1]

28 Feb 2025

PONE-D-24-49768R1Analysis of growing season drought characteristics and driving factors for vegetation in the Santun River Irrigation Area in XinjiangPLOS ONE

Dear Dr. Li,

Thank you for submitting your manuscript to PLOS ONE. After careful consideration, we feel that it has merit but does not fully meet PLOS ONE’s publication criteria as it currently stands. Therefore, we invite you to submit a revised version of the manuscript that addresses the points raised during the review process.

We look forward to receiving your revised manuscript.

Kind regards,

Nguyen-Thanh Son, Ph.D.

Academic Editor

PLOS ONE

Journal Requirements:

Reviewers' comments:

Reviewer's Responses to Questions

**Comments to the Author**

1. If the authors have adequately addressed your comments raised in a previous round of review and you feel that this manuscript is now acceptable for publication, you may indicate that here to bypass the “Comments to the Author” section, enter your conflict of interest statement in the “Confidential to Editor” section, and submit your "Accept" recommendation.

Reviewer #4: (No Response)

Reviewer #5: All comments have been addressed

Reviewer #6: All comments have been addressed

Reviewer #7: (No Response)

Reviewer #8: (No Response)

Reviewer #9: (No Response)

2. Is the manuscript technically sound, and do the data support the conclusions?

Reviewer #4: Yes

Reviewer #5: Yes

Reviewer #6: Yes

Reviewer #7: Yes

Reviewer #8: Yes

Reviewer #9: Yes

3. Has the statistical analysis been performed appropriately and rigorously? 

Reviewer #4: Yes

Reviewer #5: N/A

Reviewer #6: Yes

Reviewer #7: Yes

Reviewer #8: Yes

Reviewer #9: Yes

4. Have the authors made all data underlying the findings in their manuscript fully available?

Reviewer #4: Yes

Reviewer #5: Yes

Reviewer #6: Yes

Reviewer #7: Yes

Reviewer #8: Yes

Reviewer #9: Yes

5. Is the manuscript presented in an intelligible fashion and written in standard English?

Reviewer #4: Yes

Reviewer #5: No

Reviewer #6: Yes

Reviewer #7: Yes

Reviewer #8: Yes

Reviewer #9: Yes

6. Review Comments to the Author

Reviewer #4: The study titled “Analysis of Growing Season Drought Characteristics and Driving Factors for Vegetation in the Santun River Irrigation Area in Xinjiang” presents a remote sensing-based assessment of drought trends. The manuscript very well articulated the drought analysis in arid areas of Xinjiang. It also addressed the research applicability and drought management strategies. However, there are certain suggestions that the author should take into account:

• In the introduction section, the manuscript mentioned the study of droughts and its approaches but lacks a clear research gap and objectives that the study aims to address.

• The introduction deals more about the methods, such as TVDI, Theil-Sen trend analysis, etc., which should be appropriate for the methods section. The introduction should focus on the “how” aspect rather than “what” and "why," conveying the novelty of the research.

• Mention briefly why Santun RIA is significant for studying drought characteristics and its driving factors.

• In the methodology section, briefly mention the rationale behind choosing the mentioned methods for easy reproducibility of the work.

• The methodology section outlines various methods but does not justify why these were chosen for drought analysis.

• The study analyzes the drought conditions over a specific period, but it could be improved by exploring the implications for future water resource management and agricultural planning.

• The paper did not analyze the groundwater depletion in detail, which has been discussed as key factors for drought intensification in Xinjiang. Include the suggestion in the future research section.

• The manuscript does not address human-induced factors such as urban expansion, irrigation, and policies.

Reviewer #5: I see articles have been revised well as per comments given in previous review. The topic undertaken is needs of the time and methods applied are relevant. The article is nicely designed and written well but I suggest authors to do a comprehensive language check as at several places English language is poor and sentences are not clear. Also, discussion needs a comprehensive revision as it is lengthy and did not provide a meaningful discussion. My specific comments are as:

Kindly avoid using personal pronouns like ‘we’, ‘you’, ‘our’, ‘I’, ‘us’, etc.

Second sentence in abstract may be revised.

Keywords must be revised. The keyword ‘remote sensing monitoring’ did not make a clear meaning while the full form of TDVI should be written in keyword. Similarly, drought and Landsat are also not reliable as keywords, instead it may be written as ‘drought monitoring’ and ‘Landsat datasets’.

Line 73. Add citation.

Table 1. Kindly add a column for each Landsat satellite data used.

Table 2 may be removed and the full form of each abbreviation used should be mentioned in manuscript at first appearance.

Quality of figure 3, 9 and 10 should be improved.

Some studies have been done to monitor drought using Landsat data and temperature and vegetation indices. Authors should check these studies and compare the findings with these studies in the manuscript. Some of the studies are: https://doi.org/10.1016/j.ejrh.2024.101689,
https://doi.org/10.1007/s10661-022-10028-5,
https://doi.org/10.1016/j.ecolind.2023.110584,
https://doi.org/10.1007/978-981-19-3567-1_4,
https://doi.org/10.1016/j.ecolind.2024.112681

Discussion seems to be a summary especially sub-section 4.2-4.5. Authors should try to build arguments based on the findings of this study and comparing it with previous studies. Kindly try to provide causes and effects of major findings and add a sub-section of policy implication at the end. In current form the discussion is lengthy but did not provide a concrete discussion.

Kindly bring whole conclusion in one or two paragraph.

Reviewer #6: I have read the draft “Analysis of growing season drought characteristics and driving factors for vegetation in the Santun River Irrigation Area in Xin jiang” revised based on the comments received earlier from three reviewers.

I have found that the authors have addressed the concerns raised by the previous reviewers. The contexts and arguments placed in the paper are convincing.

Water availability to the requirements of specific crops is the prime production condition, absence of which, could nullify contributions of other inputs. There are specific thresholds of optimal water requirement for every crop, which the precipitation and/or irrigation provisioning need to ensure in case water availability deviates from optimality. Thus, there involves optimization of costs for water provisioning, and all requires precise estimates on the cost and benefits derived from land use. These aspects though are not explicit in the paper, one could relate the significance the paper posits.

The context of the study is thus presented well - a timely and accurate assessment of the drought situation (water availability assessment to the requirements of crops cultivated) is of great significance in sustaining production. The authors are well aware of the developments in the field of GIS and applications, consider a host of factors including the topography and human activities as drivers of drought.

There is high water demand in the study area, though causes of demand are indicated, could be explained further as – water availability declines because of variation of precipitation, and higher water use intensity is because of cropping pattern change driven by the market. and higher human consumption of water in new areas of consumption.

Data sources are well indicated and methodology is well explained in the paper to measure the TVDI. Geodetector model however requires detailed justification on how all the factors act as drivers. If the results show drought reduction and enhancement of drought across various seasons, the explanatory factors need detailed discussion. For instance, the main reason for the gradual increase of drought in summer is temperature and increased need of water for crops, as because it is the prime growing season of crops. Further, the study area has high population density, and the demand for crops would be market driven, and market as a factor to intensify input use to derive higher yield and even change cropping patterns for exotic crops, which may require more water. Detailed discussion on these lines along with crop data could have made the analysis comprehensive.

Drought could be linked to cropping seasons, assessing water requirements for crops cultivated, and drought could be aggravated/reduced by introduction/withdrawal of water intensive crops. Mapping of crops to the seasons could have arrived to derive better results from the exercises. There is scope to have simulation exercises on water requirement on three fronts - cropping pattern to the ecological condition of the study area; introduction of water intensive crops driven by market, and introduction of low water intensive crops driven/compensated by state subsidy; and as indicated technology can complement/aggravate the rest of the interventions to ensure optimality or aggravating the droughts too.

Reviewer #7: The research study provides a detailed, well-structured analysis of drought trends in the Xinjiang Santun River Irrigation District, integrating remote sensing and statistical models. The study effectively identifies spatial heterogeneity, emphasizing that the northern part of the irrigation area experiences more severe drought than the southern part. However, some revisions and improvements are needed in the study before publication. By addressing these suggestions, the study will meaningfully contribute to drought risk assessment, water resource management, and climate change adaptation strategies in arid agricultural regions.

1. The study categorizes drought severity but does not explain the basis and threshold values for these classifications (e.g., what threshold defines “moderate” vs. “severe” drought?)

2. There is a need to discuss the potential bias in TVDI estimates, especially in sparsely vegetated regions where NDVI approaches zero and may fail to represent soil moisture conditions.

3. The study needs to discuss the model uncertainty and how the data limitations (like- cloud cover, sensor errors) may affect the results.

4. Data on groundwater depletion needs to be incorporated to establish the claim of over-extraction.

5. The study needs to discuss how various irrigation policies and water allocation strategies implemented in the area affect drought severity.

Reviewer #8: Author need to check and use concise language to enhance readability for readers. Also needs recheck the PLOS ONE Guidelines thats why requires some minor corrections before it can be accepted for publication.

Manuscript Title: “Analysis of growing season drought characteristics and driving factors for vegetation in the Santun River Irrigation Area in Xinjiang”.

Manuscript ID: PONE-D-24-49768R1

Dear Authors,

I enjoyed reading this work. However, I have just some minor comments which are given section-wise.

The manuscript addresses a very relevant matter on the growing season drought characteristics and driving factors for vegetation. It highlights some pertinent issues in the context of agricultural production and the economic crisis of the global south region. Therefore, The topic is interesting and could be an important contribution to the journal and the discipline of climatic paradigm and agricultural production. However, the manuscript requires some minor corrections before it can be accepted for publication.

General comments:

• Use concise language to enhance readability for readers.

• Research gap needs to be justified with the help of previous literature.

• The methods section should be written clearly and comprehensively to ensure accessibility and understanding for readers. Basically who are without a background of quantitative research.

• Methodology portions needed to be written sequentially and needed to be justified with previous literature review.

• More policy recommendations need to be discussed by underpinning the findings.

Abstract:

The abstract provides an overview of the research topic but lacks of conciseness – by following the structure sequentially like background, objectives, methods, results, and conclusions would provide a standard structure of the abstract.

Introduction:

• Overall, the introduction portion is comprehensive, but the sentences lack of proper linkages.

• Therefore, the suggestion is to critically evaluate the literature to highlight specific gaps by incorporating more recent studies to reflect advancements in the climatic paradigm, including human aspects.

Study Area:

• After replacing this “Study Area, Data, and Methods” authors should write study area only in the line of 96.

• The paragraph needs to include why Xinjiang Santun River Irrigation District is more important than others in the context of climatic chapters with a proxy of drought.

Data and Methods:

• “In line 116 the title “Data sources and Processing” should be corrected as “Material and Methods”.

• In line 130 the title “Methods” should be corrected as “Methods of Data Analysis”.

• Minimal discussion has been done on how the data collection methods ensure reliability and validity. So, the suggestion is to discuss potential biases and how they are allayed in the data collection process by citing previous literature.

Results and Discussion:

To improve the quality of work, findings should be analyzed in reference to a theoretical framework. Include a comparison discussion with similar studies to contextualize the results in the same periphery.

Conclusion:

A concise summary should be included in the conclusion, along with key points from the results and discussion, while also addressing the study's limitations to provide a comprehensive closing perspective for this study.

Reviewer #9: (No Response)

7. PLOS authors have the option to publish the peer review history of their article (what does this mean? ). If published, this will include your full peer review and any attached files.

**Do you want your identity to be public for this peer review?** For information about this choice, including consent withdrawal, please see our Privacy Policy .

Reviewer #4: No

Reviewer #5: **Yes: ** Dr. Shahfahad

Reviewer #6: **Yes: ** Kalyan Das

Reviewer #7: No

Reviewer #8: No

Reviewer #9: No

---

## [Author Response · Author response to Decision Letter 2]

11 Mar 2025

Thank you for your letter. We would like to thank the reviewers and editorial department for their comments on our manuscript. We have revised the manuscript based on the reviewer's comments and are submitting it again for your approval. Our replies to the reviewers’ comments are presented below. (Note: The line numbers in “Please refer to lines .” are from the Manuscript file.)

Reviewer #4

The study titled “Analysis of Growing Season Drought Characteristics and Driving Factors for Vegetation in the Santun River Irrigation Area in Xinjiang” presents a remote sensing-based assessment of drought trends. The manuscript very well articulated the drought analysis in arid areas of Xinjiang. It also addressed the research applicability and drought management strategies. However, there are certain suggestions that the author should take into account:

1.In the introduction section, the manuscript mentioned the study of droughts and its approaches but lacks a clear research gap and objectives that the study aims to address.

Answer: Based on your suggestions, we have rewritten the introduction of the paper to enhance the description of the shortcomings of drought research in the present stage and to clearly present the research objectives of this study. In addition, we have appropriately added research on drought-driven mechanisms, making the introduction more closely related to the topic of this paper. Please refer to lines 76-109.

2.The introduction deals more about the methods, such as TVDI, Theil-Sen trend analysis, etc., which should be appropriate for the methods section. The introduction should focus on the “how” aspect rather than “what” and "why," conveying the novelty of the research.

Answer: After in-depth discussion by the team, we have optimized the introduction section of the paper. We have streamlined the detailed description of the specific indicators and methods, focusing instead on how drought hazards should be studied and the current research progress. Please refer to lines 34-64. In addition, we have supplemented the discussion of the driving mechanisms of drought to ensure that the introduction is closely related to the topic of the article. Please refer to lines 65-75. On this basis, we now clear point out the shortcomings of the existing research and the shortcomings of the research area so as to naturally lead to the core goal of this study. Through these adjustments, we aimed to better highlight the innovation and value of this research.Please refer to lines 76-109.

3.Mention briefly why Santun RIA is significant for studying drought characteristics and its driving factors.

Answer: I have thought deeply about the questions you have posed and respond to them in detail here. The Santun River Irrigation District in Xinjiang is important for the study of drought characteristics and its drivers. The main reasons for this are presented below.

First, the Santun River Irrigation District in Xinjiang is located in the arid zone on the northern slopes of the Tianshan Mountains where water resources are scarce and the ecosystems are fragile, and the impact of drought on agricultural production and the ecological environment is significant. In addition, the discussion section of this paper also points out that the land use types in this area have changed significantly in the last 20 years, and this nonlinear interaction between natural and human factors provides an ideal case for analyzing the drought-driving mechanism.

Second, as the core area of the agricultural economy in Changji City, Xinjiang, the Santun River Irrigation District in Xinjiang is mainly dominated by high water-consuming crops, such as cotton and wheat, and studying the spatial and temporal differentiation patterns of drought in this region can reveal the response threshold of oasis agricultural systems to climate change and provide a scientific basis for optimizing the irrigation system and planting structure.

Finally, most existing studies focused on large-scale regions (e.g., the North China Plain and the Tibetan Plateau in China), and there is a relative lack of detailed studies on the north slope of the Tianshan Mountains, which is a key area of ecological-economic synergy. The study of the Santun River Irrigation District can fill the regional gap in drought-driving research and provide a reference for sustainable development in similar arid zones.

In summary, the Santun River Irrigation Area in Xinjiang is of great significance for studying drought characteristics and its driving factors, and it is a very good representative area. To prevent readers from having doubts, based on your suggestions, we have pointed out the importance of the research area and the existing research gaps, as well as the research objectives of this study, in the introduction of the paper. Please refer to lines 87-91.

4.In the methodology section, briefly mention the rationale behind choosing the mentioned methods for easy reproducibility of the work.

Answer: Thank you for your review. A rationale for the selection of each method has been added to the research methods section of this paper. Please refer to lines 171-175, 189-193, 212-215, 221-225.

5.The methodology section outlines various methods but does not justify why these were chosen for drought analysis.

Answer: Based on your suggestion, the reasons for choosing each method have been added to the research methods section of this article. Please refer to lines 171-175, 189-193, 212-215, 221-225.

6.The study analyzes the drought conditions over a specific period, but it could be improved by exploring the implications for future water resource management and agricultural planning.

Answer: We have reworked the section on coping strategies in the discussion section of the paper based on your suggestions. Please refer to lines 565-582.

7.The paper did not analyze the groundwater depletion in detail, which has been discussed as key factors for drought intensification in Xinjiang. Include the suggestion in the future research section.

Answer: Thank you for your advice. The importance of groundwater depletion factors has been added to subsection 4.4 of the discussion section. Please refer to lines 604-606.

8.The manuscript does not address human-induced factors such as urban expansion, irrigation, and policies.

Answer: Based on your suggestions, we have revised the discussion section to deepen the discussion of the influences of human factors on drought. In addition, in subsection 4.1 of the discussion section, we discuss the effects of human activities (e.g., changes in land use types and urban expansion) on drought. Appropriate reference is now made to the effects of drought on irrigation systems and groundwater extraction.Please refer to lines 518-536; .

In addition, in the analysis of the driving factors in the results section, we also considered human activities (land use type change and gross domestic product) and included them in the analysis of the driving factors. However, the effect of human activities is not as great as the influence of the natural factors on the drought. Please refer to lines 447-480.

Reviewer #5

I see articles have been revised well as per comments given in previous review. The topic undertaken is needs of the time and methods applied are relevant. The article is nicely designed and written well but I suggest authors to do a comprehensive language check as at several places English language is poor and sentences are not clear. Also, discussion needs a comprehensive revision as it is lengthy and did not provide a meaningful discussion. My specific comments are as:

1.Kindly avoid using personal pronouns like ‘we’, ‘you’, ‘our’, ‘I’, ‘us’, etc.

Answer: Thank you for the suggestions, we have revised the paper as a whole with reference to the comments of the reviewers and have enhanced the logic and accuracy of the language of the article. We have also hired a professional language editing service to thoroughly check and touch up the manuscript.

2.Second sentence in abstract may be revised.

Answer: We have revised the abstract of the paper in strict accordance with the "background, purpose, methods, results, and conclusion" framework for the summary part of the writing. The description of the methods in the second sentence of the abstract has been simplified to balance the overall structure of the abstract. Please refer to lines 14-30.

3.Keywords must be revised. The keyword ‘remote sensing monitoring’ did not make a clear meaning while the full form of TDVI should be written in keyword. Similarly, drought and Landsat are also not reliable as keywords, instead it may be written as ‘drought monitoring’ and ‘Landsat datasets’.

Answer: This has been modified accordingly.Please refer to lines 31-32.

4.Line 73. Add citation.

Answer: We have reorganized and rewritten the introduction section of the article. Where relevant citations needed to be added, we have rechecked and added them. Please refer to lines 34-109.

5.Table 1. Kindly add a column for each Landsat satellite data used.

Answer: As there are too many remote sensing images involved in this paper, this information cannot be clearly expressed in Table 1. After discussion among the team, we decided to organize each data image information used into a remote sensing image information table, which is convenient for readers to read and understand. The detailed information is presented in Table 2. In addition, sensor models designed in the image are also noted in Table 2. Please refer to lines 154-157.

6.Table 2 may be removed and the full form of each abbreviation used should be mentioned in manuscript at first appearance.

Answer: According to your suggestions, we have removed this table and define each acronym the first time it is used in the article.

7.Quality of figure 3, 9 and 10 should be improved.

Answer: Based on your suggestions, we have redrawn Figures 3, 9, and 10 in the paper. We added additional boundary conditions and improved the quality of the images. Please refer to lines 268, 374, 377.

8.Some studies have been done to monitor drought using Landsat data and temperature and vegetation indices. Authors should check these studies and compare the findings with these studies in the manuscript. Some of the studies are:

https://doi.org/10.1016/j.ejrh.2024.101689,
https://doi.org/10.1007/s10661-022-10028-5,
https://doi.org/10.1016/j.ecolind.2023.110584,
https://doi.org/10.1007/978-981-19-3567-1_4,
https://doi.org/10.1016/j.ecolind.2024.112681

Discussion seems to be a summary especially sub-section 4.2-4.5. Authors should try to build arguments based on the findings of this study and comparing it with previous studies. Kindly try to provide causes and effects of major findings and add a sub-section of policy implication at the end. In current form the discussion is lengthy but did not provide a concrete discussion.

Answer: Regarding the several research papers you mentioned, I have carefully read them and have referenced them and similar studies in the discussion section of this paper to support the research results and discussion. Please refer to lines 491, 496, 505, 509, 519, 525, 535, 536, 542, 549, 558, 561.

After discussion by the team, we decided to carry out block discussion according to the paper’s framework. This may make it easier for the reader to read and understand. In addition, we have made specific arguments based on the findings and similar studies in each section of the discussion. The discussion on the causes and effects of the main research findings is scattered in each section and mainly analyzes the reasons and driving mechanisms behind this phenomenon. Please refer to lines 488-582.

In addition, based on your suggestions, we provide policy recommendations and recommend drought response strategies in Section 4.3. to make it easier for readers to read and understand. Please refer to lines 564-582.

We have fine-tuned the discussion by reducing unnecessary narrative and refining the language. We hired a professional language editing service to thoroughly check and polish the manuscript.

9.Kindly bring whole conclusion in one or two paragraph.

Answer: Thank you for your suggestion. After discussion by the team, we decided to summarize the research according to three points, which may make it more convenient for readers to read and understand. In addition, according to your suggestion, we have simplified the conclusion and summarized the main conclusions it in concise language. In addition, in the last paragraph, we have also added appropriate statements on policy recommendations in accordance with other reviewers' comments to provide an overall conclusion to the article.Please refer to lines 620-643.

Reviewer #6

I have read the draft “Analysis of growing season drought characteristics and driving factors for vegetation in the Santun River Irrigation Area in Xin jiang” revised based on the comments received earlier from three reviewers. I have found that the authors have addressed the concerns raised by the previous reviewers. The contexts and arguments placed in the paper are convincing. Water availability to the requirements of specific crops is the prime production condition, absence of which, could nullify contributions of other inputs. There are specific thresholds of optimal water requirement for every crop, which the precipitation and/or irrigation provisioning need to ensure in case water availability deviates from optimality. Thus, there involves optimization of costs for water provisioning, and all requires precise estimates on the cost and benefits derived from land use. These aspects though are not explicit in the paper, one could relate the significance the paper posits. The context of the study is thus presented well - a timely and accurate assessment of the drought situation (water availability assessment to the requirements of crops cultivated) is of great significance in sustaining production. The authors are well aware of the developments in the field of GIS and applications, consider a host of factors including the topography and human activities as drivers of drought. There is high water demand in the study area, though causes of demand are indicated, could be explained further as – water availability declines because of variation of precipitation, and higher water use intensity is because of cropping pattern change driven by the market. and higher human consumption of water in new areas of consumption.

Data sources are well indicated and methodology is well explained in the paper to measure the TVDI. Geodetector model however requires detailed justification on how all the factors act as drivers. If the results show drought reduction and enhancement of drought across various seasons, the explanatory factors need detailed discussion. For instance, the main reason for the gradual increase of drought in summer is temperature and increased need of water for crops, as because it is the prime growing season of crops. Further, the study area has high population density, and the demand for crops would be market driven, and market as a factor to intensify input use to derive higher yield and even change cropping patterns for exotic crops, which may require more water. Detailed discussion on these lines along with crop data could have made the analysis comprehensive.

Drought could be linked to cropping seasons, assessing water requirements for crops cultivated, and drought could be aggravated/reduced by introduction/withdrawal of water intensive crops. Mapping of crops to the seasons could have arrived to derive better results from the exercises. There is scope to have simulation exercises on water requirement on three fronts - cropping pattern to the ecological condition of the study area; introduction of water intensive crops driven by market, and introduction of low water intensive crops driven/compensated by state subsidy; and as indicated technology can complement/aggravate the rest of the interventions to ensure optimality or aggravating the droughts too.

Answer: Thank you for your suggestions.

First, regarding your comment that "The Geodetic model needs to specify how all the factors act as drivers," in Section 2.3.4 of the paper, we added tables to explain the driving logic of the different dri

---

## [Decision Letter · Decision Letter 2]

17 Apr 2025

Analysis of growing season drought characteristics and driving factors for vegetation in the Santun River Irrigation Area in Xinjiang

PONE-D-24-49768R2

Dear Dr. Qiao Li,

We’re pleased to inform you that your manuscript has been judged scientifically suitable for publication and will be formally accepted for publication once it meets all outstanding technical requirements.

Kind regards,

Nguyen-Thanh Son, Ph.D.

Academic Editor

PLOS ONE

Reviewer #4: The authors have successfully incorporated all the suggestions I provided. Therefore, I accept the paper for publication in its current form.

Reviewer #5: Authors have revised the MS as per comments. I feel manuscript has been improved significantly and may be recommended for publication now.

Reviewer #6: The authors have addressed all comments explicitly, or implicitly. To address certain questions it is understood that there are data gaps. The authors in such contexts, have led detailed discussion on the concerns raised.

Reviewer #7: The manuscript titled "Analysis of Growing Season Drought Characteristics and Driving Factors for Vegetation in the Santun River Irrigation Area in Xinjiang" presented a comprehensive and insightful analysis of the factors influencing vegetation response to drought conditions in a critical irrigation area. The authors have addressed all the queries raised during the review process and have provided adequate explanations and revisions to enhance the clarity and depth of their analysis. The methodology is robust, and the findings are well-supported by data. The manuscript now meets the standards required for publication and provides valuable information for future research and practical applications in drought management.

I recommend the paper for acceptance in its current form.

---

## [Editor Report · Acceptance letter]

PONE-D-24-49768R2

PLOS ONE

Dear Dr. Li,

I'm pleased to inform you that your manuscript has been deemed suitable for publication in PLOS ONE. Congratulations! Your manuscript is now being handed over to our production team.

Kind regards,

on behalf of

Dr. Nguyen-Thanh Son

Academic Editor

PLOS ONE